# DiffusionVLA: Scaling Robot Foundation Models via Unified Diffusion and Autoregression

**Junjie Wen** [1 2 *] **Yichen Zhu** [1 * †] **Minjie Zhu** [1 2 *] **Zhibin Tang** [1] **Jinming Li** [3]
**Zhongyi Zhou** [2] **Xiaoyu Liu** [3] **Chaomin Shen** [2] **Yaxin Peng** [3] **Feifei Feng** [1]

**diffusion-vla.github.io**

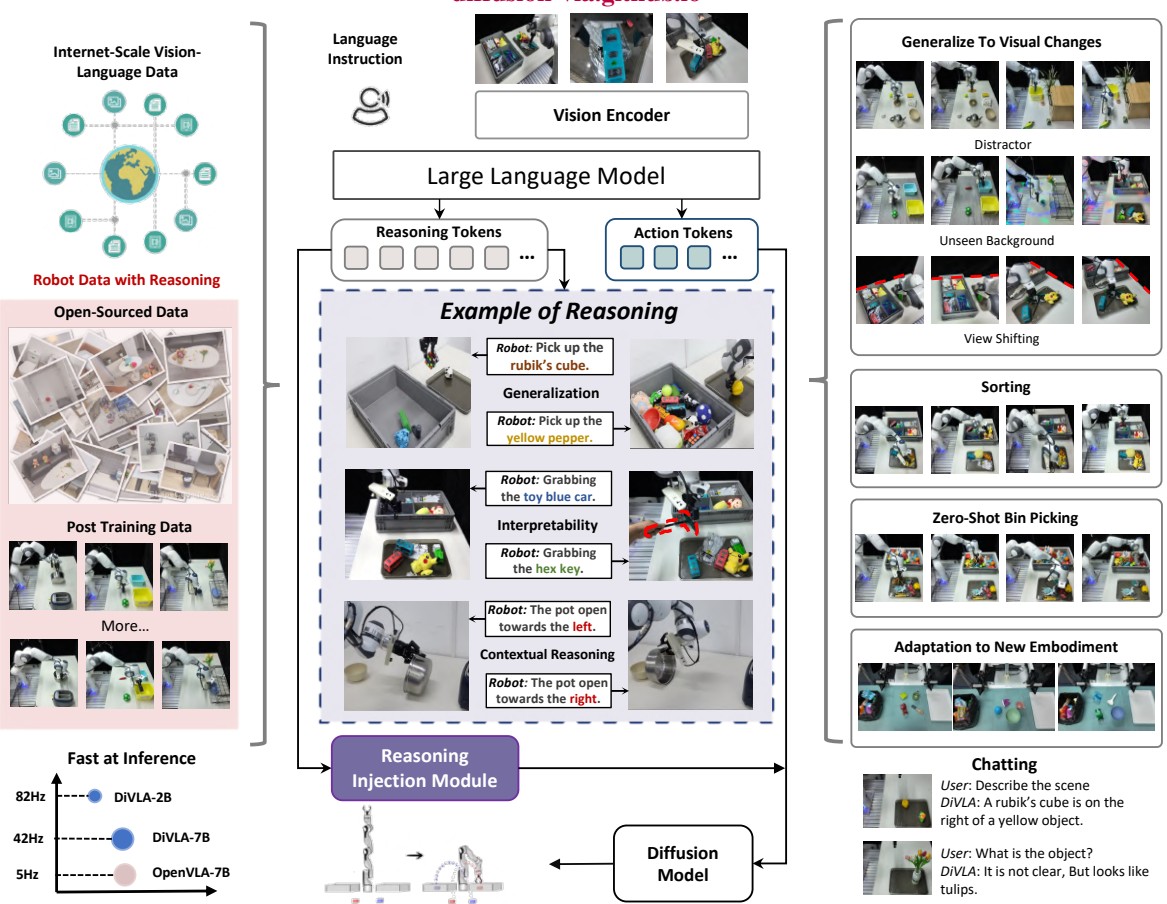

Figure 1: Our proposed DiffusionVLA model unifies autoregressive and diffusion modeling to enable self-reasoning and robot policy learning. This approach generalizes effectively to visual changes, supports zero-shot bin picking, adapts to new robot morphologies, performs visual question-answering, and generates actions with high speed.

## Abstract

In this paper, we present DiffusionVLA, a novel framework that integrates autoregressive reasoning with diffusion policies to address the limitations of existing methods: while autoregressive Vision-Language-Action (VLA) models lack precise and robust action generation, diffusion-based policies inherently lack reasoning capabilities. Central to our approach is autoregressive reasoning — a task decomposition and explanation process enabled by a pre-trained VLM — to guide diffusion-based action policies. To tightly couple reasoning with action generation, we introduce a reasoning injection module that directly embeds self-generated reasoning phrases into the

---

*Equal contribution, work done during Junjie Wen and Minjie Zhu's internship at Midea Group. [1]Midea Group, Shanghai, China [2]East China Normal University, Shanghai, China [3]Shanghai University, Shanghai, China. Correspondence to: Yichen Zhu <zhuyc25@midea.com>.

*Proceedings of the 42nd International Conference on Machine Learning*, Vancouver, Canada. PMLR 267, 2025. Copyright 2025 by the author(s).

policy learning process. The framework is simple, flexible, and efficient, enabling seamless deployment across diverse robotic platforms.

We conduct extensive experiments using multiple real robots to validate the effectiveness of DiVLA. Our tests include a challenging factory sorting task, where DiVLA successfully categorizes objects, including those not seen during training. The reasoning injection module enhances interpretability, enabling explicit failure diagnosis by visualizing the model's decision process. Additionally, we test DiVLA on a zero-shot bin-picking task, achieving **63.7% accuracy on 102 previously unseen objects**. Our method demonstrates robustness to visual changes, such as distractors and new backgrounds, and easily adapts to new embodiments. Furthermore, DiVLA can follow novel instructions and retain conversational ability. Notably, DiVLA is data-efficient and fast at inference; our smallest DiVLA-2B runs 82Hz on a single A6000 GPU. Finally, we scale the model from 2B to 72B parameters, showcasing improved generalization capabilities with increased model size.

## 1. Introduction

Recently, Vision-Language-Action (VLA) models have emerged as a promising direction in robotics (Bu et al., 2024; Liu et al., 2024b; Li et al., 2024c;a; Zhang et al., 2024; Zheng et al., 2025; Pertsch et al., 2025; Brohan et al., 2023; Kim et al.; Wen et al., 2024; Black et al., 2024; Octo Model Team et al., 2024; Deng et al., 2025; Liu et al., 2025; Cui et al., 2025; Chen et al., 2025; Liu et al., 2024a). Among these VLAs, a prominent approach frames action prediction as a next-token prediction (NTP) task, mirroring the dominant autoregressive paradigm in Large Language Models (LLMs), which operates by sequentially predicting discrete tokens, with each token's generation conditioned on the preceding ones. While these models, such as RT-2 (Brohan et al., 2023) and OpenVLA (Kim et al.), have demonstrated notable success, they suffer from inherent limitations. First, discretizing continuous action data into fixed-size tokens can disrupt the coherence and precision of actions. Second, NTP is inherently inefficient for action generation, particularly in real-time robotic applications where performance is critical.

Meanwhile, building on the success of diffusion models in content generation (Rombach et al., 2022; Peebles & Xie, 2023; Podell et al., 2023; Ho et al., 2020; 2022), diffusion-based models for learning visuomotor policies (Chi et al., 2023) have gained significant popularity over the past two years. Numerous methods have demonstrated strong perfor-

mance in manipulation tasks by modeling action sequence generation as a noise-denoising process. This approach better captures the multimodal nature of robotic actions and enables faster sequence generation compared to NTP-based VLA models. However, despite the advantages of diffusion models for policy learning, they lack the reasoning capabilities crucial for VLA models to solve complex tasks effectively, a component that evidently improves LLMs. This motivates us to raise the question: *can we bring together the advantages of both parties, specifically the reasoning power of autoregressive models and the robustness of high-frequency action generation offered by diffusion models?*

In this work, we propose a unified model, named ***Diffusion-VLA (DiVLA in short)***, that integrates autoregression with a diffusion model. The autoregressive component provides reasoning over the query, while the diffusion model controls the robot. Specifically, DiVLA builds upon a pre-trained Vision-Language Model (VLM), retaining its autoregressive capabilities for text-based reasoning. We extend this foundation by integrating a diffusion model that facilitates learning robotic actions through a noise-denoising process. This setup empowers DiVLA to achieve both language-driven reasoning and robust action generation in robotic contexts. However, simply combining these elements does not fully exploit the reasoning potential, as there is often an implicit gap between logical reasoning and actionable robot policies. To bridge this gap, we propose a reasoning injection module, which reuses reasoning outputs and embeds them directly into the policy head, thus enriching the policy learning process with explicit reasoning signals. This innovation allows us to directly incorporate reasoning into action generation, enhancing the model's dexterity, robustness, and generalization across various scenarios. Our experiments confirm that DiVLA delivers the following advantages: 1) **Visual generalization via self-generated reasoning**: DiVLA can recognize and categorize previously unseen objects via self-generated reasoning, showcasing its ability to generalize to novel visual inputs.

2) **Strong action interpretability**: Our reasoning injection module provides insights into the end-to-end policy's decision-making, explaining robot actions and facilitating failure analysis.

3) **Adaptability to novel instructions and conversational capability**: Our approach can execute novel instructions while maintaining conversational fluency, offering a versatile response range in interactive scenarios.

4) **Fast adaptation to other embodiment**: DiVLA can quickly fine-tune for deployment on new embodiment, such as bimanual robots, achieving high performance with minimal adjustments.

5) **Fast inference speed**: With inference rates of 82Hz for DiVLA-2B and 42Hz for DiVLA-7B on a single A6000 GPU, our method ensures real-time responsiveness even in

high-demand environments.

6) **Enhanced visual generalization**: DiVLA is unaffected by visual distractions or novel backgrounds, demonstrating robustness in visually dynamic settings.

7) **Scalability**: The scalable DiVLA family (2B, 7B, and 72B) demonstrates that generalization and performance improve with model size, consistent with established scaling laws.

This work introduces a novel end-to-end framework for robot policy learning. By unifying an autoregressive objective with a diffusion model, our approach enables the model to both "talk" and "act." Critically, our proposed reasoning injection module promotes generalization to novel objects and provides interpretable actions. While the concept is simple and straightforward, the combination of ingredients and the application to robot learning is novel. Empirical evaluation across multiple complex real-world robot tasks demonstrates a level of generality surpassing that of previous single-model approaches.

## 2. Related Works

**Autoregression models.** Predicting the next token has been regarded as a key approach toward general artificial intelligence due to its success in training language models (Touvron et al., 2023a;b; Achiam et al., 2023; Bi et al., 2024; Team, 2024). RT-2 (Brohan et al., 2023) pioneered the application of next-token prediction in robot learning which predicts actions by converting continuous actions into discrete tokens for learning robotic motion. Building on this, OpenVLA (Kim et al.) introduced an open-source, improved, and smaller version of RT-2 (Brohan et al., 2023) with a similar approach, while ECoT (Zawalski et al., 2024) developed a chain-of-thought method. However, research has shown that next-token prediction may not be the optimal approach for robot models, especially when adapting to various embodiments. In this work, we leverage the strength of next-token prediction specifically for reasoning tasks.

**Diffusion models.** Diffusion models have become dominant in the field of visual generation. The Diffusion Policy (Chi et al., 2023) extends the application of diffusion models to robot learning, demonstrating their effectiveness in handling multimodal action distributions. Subsequent work has advanced the Diffusion Policy (Zhao et al.; Wang et al., 2024d; Prasad et al., 2024; Reuss et al., 2024; Uehara et al., 2024a;b; Black et al., 2023a;b; Dasari et al., 2024; Lin et al., 2024) by applying it to 3D settings (Ke et al., 2024; Ze et al., 2024b;a; Yan et al., 2024), scaling it up (Zhu et al., 2024), increasing its efficiency (Jia et al.; Wang et al., 2024e), and introducing architectural innovations. For instance, TinyVLA (Wen et al., 2024) integrates diffusion models with lightweight vision-language models, while $\pi_0$ (Black et al., 2024) leverages flow matching

rather than diffusion for action generation. Our approach introduces reasoning—a crucial element in language models—into the diffusion-based vision-language-action (VLA) model.

**Robot foundation models.** Existing works (Belkhale et al., 2024; Brohan et al., 2022; 2023; Gu et al., 2023; Jiang et al., 2022; Jang et al., 2022; Fu et al., 2024; Huang et al.) leverage RL (Yuan et al., 2024) and LLM (Liang et al., 2023) to decouple the multimodal understanding and embodied control. Another line of research leverages pre-trained Vision-Language Models (VLMs) and fine-tunes them directly on robotic data (Brohan et al., 2023; Zawalski et al., 2024; Kim et al.; Wen et al., 2024). Our work follows this approach, unifying autoregressive and diffusion models for both reasoning and manipulation tasks.

**Unified auto-regressive model and image generation.** Recent work has focused on unifying multimodal understanding with image generation. These efforts include using autoregressive methods to generate images (Li et al., 2024b; Sun et al., 2024; Wang et al., 2024c;a), diffusion models to produce text, or combining both approaches into unified models (Ge et al., 2024; Lu et al., 2024; Zhao et al., 2024; Wu et al., 2024a), such as Show-O (Xie et al., 2024), Transfusion (Zhou et al., 2024), and Vila-U (Wu et al., 2024b). Unlike these methods, our framework explores unifying next-token prediction with diffusion models, which enhances the robot model's reasoning abilities. This reasoning capability, in turn, improves the model's generalization to various tasks and environments.

## 3. Methodology

Our ultimate goal is to create a unified framework that combines autoregressive models, which excel at predicting language sequences for reasoning, with diffusion models, which are highly effective at generating robotic actions. Developing such an integrated model presents substantial challenges, with key issues centered on: (i) designing an architecture that seamlessly and efficiently integrates both autoregressive and diffusion mechanisms; and (ii) leveraging self-generated reasoning to enhance action generation without adding inference computation overhead. In this section, we introduce the overall framework of our method in Section 3.1 and explore the design choices that inform our model architecture in Section 3.2.

### 3.1. Architecture

Given any sequence of interleaved images, text, and video, we first encode the images into dense visual features using SigLIP (Zhai et al., 2023). These encodings are then transformed into a fixed number of $N$ visual embeddings through a Transformer. It's worth noting that typical visual inputs

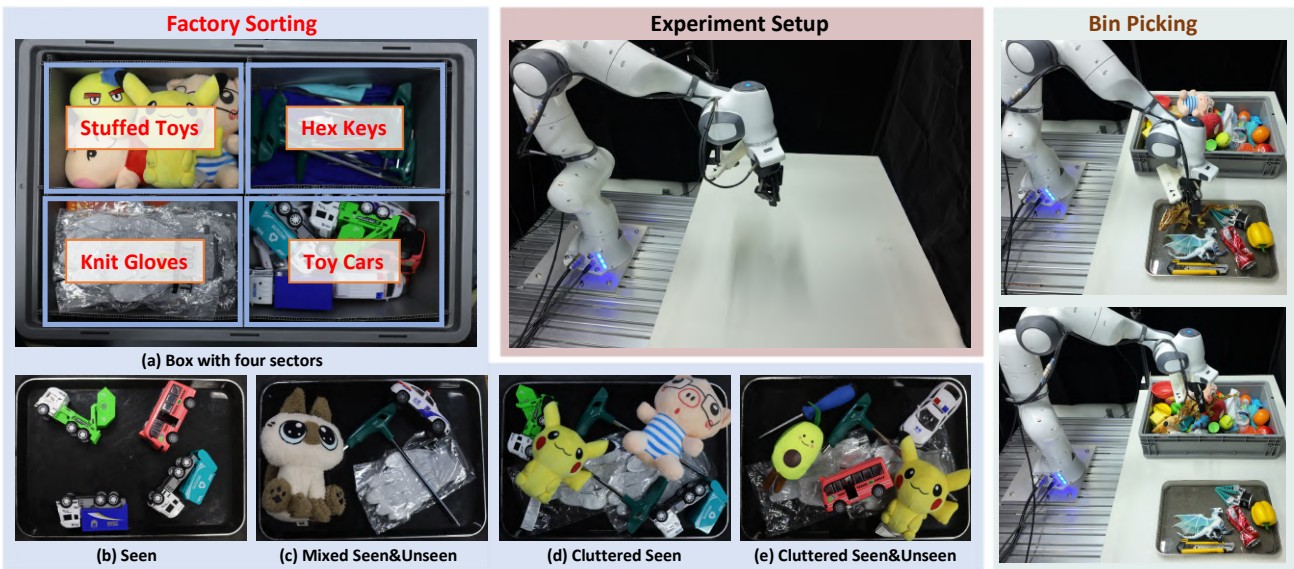

Figure 2: **Environmental Setup for the Franka Robot and Experimental Configuration for Factory Sorting. Left:** For factor sorting tasks, (a) The target sorting box is divided into four distinct sectors, each designated for one of the following categories: stuffed toys, hex keys, knit gloves, and toy cars, (c) The seen objects in the train data, (d) mixing the seen and unseen object for evaluation, (e) cluttered scene for seen objects, (f) cluttered scene for mixing seen and unseen objects. **Middle:** We use a Franka robot arm equipped with two external Zed cameras and a Realsense 435i wrist camera. **Right:** The setup for zero-shot bin picking.

Table 1: **Experimental Results for Multi-Task Learning on Real Robot.** We report the count of pre-trained trajectories. We also report the average success rate for evaluation on both in-distribution and out-of-distribution. Task 1: Select the appropriate object based on the user's intent. Task 2: Flip the vertically placed pot. Task 3: Pick up the cube and place it into the [yellow/blue] box. Task 4: Place the cup onto the plate. Task 5: Place the cube into the box.

| Model \ Tasks | Pre-trained Trajectory | In-Distribution | | | | | | Visual Generalization | | | | | |
| | | Task 1 | Task 2 | Task 3 | Task 4 | Task 5 | Avg. | Task 1 | Task 2 | Task 3 | Task 4 | Task 5 | Avg. |
| --- | --- | --- | --- | --- | --- | --- | --- | --- | --- | --- | --- | --- | --- |
| Diffusion Policy (Chi et al., 2023) | - | 66.7 | 36.4 | 0 | 36.4 | 0 | 27.9 | 11.1 | 11.1 | 0 | 22.2 | 0 | 8.9 |
| TinyVLA (Wen et al., 2024) | - | 72.7 | 45.5 | 36.4 | 45.5 | 27.3 | 45.5 | 44.4 | 44.4 | 11.1 | 22.2 | 22.2 | 28.9 |
| Octo (Octo Model Team et al., 2024) | 970K | 57.6 | 27.3 | 9.1 | 0 | 27.3 | 24.3 | 44.4 | 11.1 | 11.1 | 0 | 22.2 | 17.8 |
| OpenVLA-7B (Kim et al.) | 970K | 69.7 | 18.2 | 18.2 | 36.4 | 54.5 | 39.4 | 55.6 | 11.1 | 0 | 33.3 | 33.3 | 26.7 |
| **DiVLA-2B** | 39K | **100** | **100** | **63.6** | **63.6** | **90.9** | **83.6** | **44.4** | **66.7** | **44.4** | **66.7** | **66.7** | **57.8** |

in robot learning often include multiple camera views. To manage this, we applied the shared SigLIP visual backbone to each view and subsequently concatenated the resulting visual tokens.

For vision-language processing, we leveraged Qwen2-VL (Wang et al., 2024b), a state-of-the-art vision-language model available in three sizes: 2B, 8B, and 72B parameters. We initialized the VLM backbone with the publicly released checkpoint. It is also possible to use any other pre-trained VLM as backbone, since we decouple the vision-language understanding with action generation, making the overall architecture flexible to fit for advanced new models.

**Projection layer for action tokens**. Following the final embedding layer of the VLM, a fixed number of action tokens are generated. These tokens are then fed into a projection module, comprised of two MLP layers with

LayerNorm. This projection module functions similarly to those found in conventional vision-language models like LLaVA (Liu et al., 2023b;a), bridging the VLM's output embedding to the diffusion model and aligning their output dimensions. The diffusion model itself adheres to the standard Diffusion Policy design (Chi et al., 2023), with randomly initialized weights This component also incorporates reasoning from the LLM, which we describe in detail below. An MLP layer is attached to the last layer at the bottom of the action decoder to predict the robot's joint space. If multiple embodiments are evolved, instead of making a copy of a separate action decoder (Octo Model Team et al., 2024), we simply initialized a new MLP layer for training and evaluation. This step ensures that the knowledge from the pre-trained data is preserved and thus can quickly adapt to a new embodiment.

**Reasoning injection module.** The core of our approach

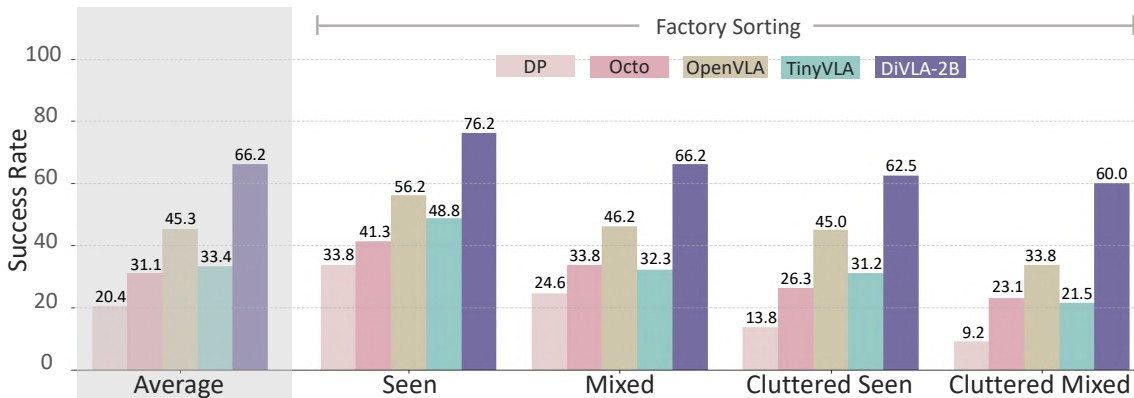

Figure 3: **Experimental Results for Factory Sorting.** We compared our DiVLA with Diffusion Policy, Octo, TinyVLA, and OpenVLA. DiVLA achieves the highest average success rate, outperforming the runner-up OpenVLA by 20.9%.

lies in introducing explicit reasoning into Vision-Language-Action (VLA) models. Unlike most autoregressive VLAs, which require a recursive setup — converting reasoning outputs into inputs for subsequent model runs — our method proposes a more efficient and streamlined integration of reasoning. By embedding reasoning directly within the policy model, we avoid the computational and operational complexities of iterative input-output cycles, enabling faster and more seamless reasoning incorporation.

Our reasoning injection module operates by taking the final embedding from the tokenized output of the reasoning component and directly injecting it into the policy model through Feature-wise Linear Modulation (FiLM) (Perez et al., 2018). This injection technique, inspired by methods in RT-1 (Brohan et al., 2022) and YAY (Shi et al., 2024), enables us to modulate the policy network's layers based on the reasoning signal. We refer to this process as "injection" because, in our design, the policy network focuses primarily on action-specific tokens, while the reasoning module functions as an auxiliary enhancement, providing contextual depth without dominating the primary decision-making flow. This approach ensures that reasoning is not only present but actively utilized during policy model training.

### 3.2. Model Design Choices

We illustrate the training strategy and other techniques that we used to improve the efficiency and effectiveness of our method. We also discuss the selection of pretraining data.

**Training objectives.** Given a batch of input sequences, the overall training loss is formulated as a combination of the diffusion loss and the next-token prediction (ntp) loss: $L = L_{diff} + \alpha L_{ntp}$, where $\alpha$ is the hyper-parameters weighting the loss term $L_{ntp}$. In our observations, the magnitude of $L_{ntp}$ consistently remains about ten times smaller than that of $L_{diff}$. To balance the contribution of each component to the overall loss, we typically set $\alpha = 10$ in all experiments. This adjustment ensures that both terms are comparably

weighted in the training process, allowing the model to learn effectively from both action and next-token prediction tasks.

**Pretraining Data.** We consider OXE (O'Neill et al., 2023) and Droid (Khazatsky et al., 2024) dataset for pretraining. We use Droid data to pre-train DiVLA-2B and DiVLA-7B. Because larger models typically needs more data for training, we use OXE and Droid together for pre-training DiVLA-72B. The original Droid data contains only robotic actions, paired partially with observations and language instructions. These data contain only robotic actions, paired partially with observations and language instructions. To enhance our model's ability to generalize with language, we leverage GPT-4o to automatically transform these data into a form that includes reasoning. Consequently, the network architecture remains consistent across both the pre-training and fine-tuning stages.

## 4. Experiments

In this section, we examine the effectiveness of DiVLA for embodied control. In Section 4.2, we compare DiVLA against other state-of-the-art models within a standard multi-task setting, assessing its performance in both in-distribution and out-of-distribution scenarios. In Section 4.3, we evaluate DiVLA in the challenging factory sorting task, showcasing its remarkable performance and illustrating how reasoning enables the model to analyze robot actions and sort items accurately. In Section 4.5, we showcase DiVLA's impressive generalization abilities in a zero-shot bin-picking task involving over 102 unseen objects. In Section 4.6, we illustrate DiVLA's adaptability to new embodiments, successfully completing complex tasks that require bimanual coordination. We provide experimental setup and implementation details in the Appendix.

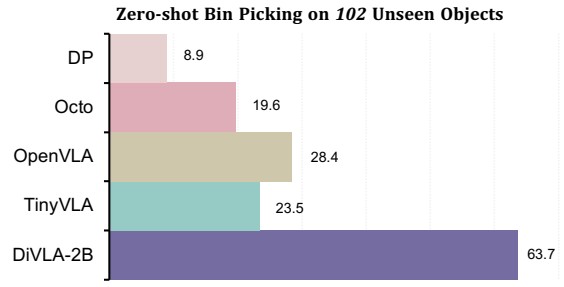

Figure 4: **Zero-shot Bin Picking on 102 Unseen Objects.** Our method outperforms the state-of-the-art robot foundation models by a large margin.

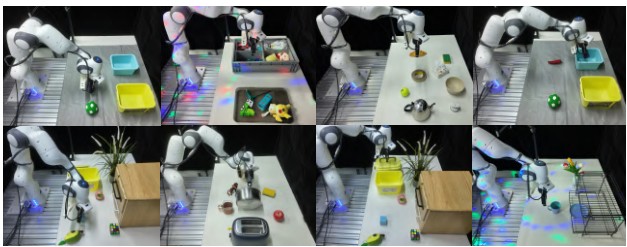

Figure 5: Examples of visual variations, including randomly placed distractors, different backgrounds, and distracting lighting. Di-VLA is robust to visual changes in different scenarios.

### 4.1. Experimental Setup.

**Implementation details and pretraiend data.** The model is pre-trained on the Droid (Khazatsky et al., 2024) dataset. We then finetune our model on evaluation tasks, similar to the setting as $\pi_0$ (Black et al., 2024). We use LoRA (Hu et al., 2021) to fine-tune the VLM models. We use 2e-5 as a fixed learning rate to train the model, similar to OpenVLA. The visual encoder and VLM are frozen. We apply LoRA on VLM for fine-tuning.

**Data for finetuning.** We explore four experimental settings: factory sorting, bin picking, multi-task learning, and table bussing. The first three settings are conducted with the Franka robot, while the table bussing task utilizes the bimanual AgileX robot. Our dataset includes 500 trajectories for the factory sorting task and 580 trajectories for multi-task learning. The bin picking task is designed as a zero-shot task, so no training data was collected for it. For the table bussing task, we gathered 400 trajectories, where objects are randomly placed on the table, often overlapping with each other. During the fine-tuning stage, all data corresponding to the same embodiment are trained together. For instance, the factory sorting and multi-task data are combined for training purposes.

### 4.2. Real-World Multi-Task Learning

We begin with a standard setting in which the model is trained on multiple tasks and completes each task based on different user queries. We designed five tasks: object selection, flip the vertically placed pot, placing a cube into a designated box, placing a cup onto a plate, and placing a cube inside a box. Detailed descriptions of these tasks are provided in the Appendix. The experimental results can be found in Table 1. We compare our method to the Diffusion Policy (Chi et al., 2023), TinyVLA (Wen et al., 2024), Octo (Octo Model Team et al., 2024), and OpenVLA (Kim et al.). Notice that both Octo and OpenVLA is pre-trained on OXE (O'Neill et al., 2023), which is 25 times larger than our pre-trained datasets.

**Generalization to visual changes.** We further evaluate our method in a multi-task setting with visual changes to assess its robustness and adaptability in diverse, dynamic environments. Specifically, we introduce three challenging scenarios designed to test the model's ability to handle visual variability: 1) adding additional distractors in the surroundings to increase visual clutter and complexity, 2) altering the background to test resilience against shifts in scene context, and 3) implementing colorful lighting effects to introduce varied illumination and color hues. These scenarios are shown in Figure 5 to illustrate the impact of each change on the visual environment. The experimental results are demonstrated in Table 1.

Our evaluation of these scenarios reveals that while all methods experience a decline in performance due to these visual changes, our method consistently maintains the highest average success rate across five different tasks. This outcome highlights the model's inherent robustness and adaptability, despite the absence of any specific data augmentation techniques during training.

### 4.3. End-to-End Sorting on Real Robot

We evaluate the capability of DiVLA in an industrial setting, where a robot is tasked with sorting items into designated sectors within a large box based on object category. Specifically, we categorize items into four classes: (1) toy cars, (2) knit gloves, (3) stuffed toys, and (4) hex keys. The language instruction provided is "Sort all items into corresponding areas". A total of 500 trajectories are collected as training data. The task is considered successful only if the robot successfully grasps the object and places it in the correct sector. The experimental setup is illustrated in Figure 2.

This task poses several challenges, requiring both precise object grasping and accurate category identification. We assess our method under two difficulty settings: easy and hard. In easy mode, fewer than 5 items are placed on the table, whereas in hard mode, 6 to 11 items are randomly arranged.

Furthermore, both seen and unseen objects are mixed in these scenarios. In the cluttered scene, items may overlap or be randomly distributed across the table, increasing the complexity of the sorting task.

The experimental results are illustrated in Figure 3. Di-VLA demonstrates robust performance with an average success rate of 66.2% across all experimental settings. While other methods show significant performance degradation as scene complexity increases (i.e., higher object count and clutter level), particularly evident in DP's sharp decline to 9.2% success rate in highly cluttered mixed scenarios, DiVLA maintains a substantial 60% success rate. This sustained performance underscores our approach's capability to effectively handle complex and dynamic real-world scenarios.

## 4.4. Behavior Analysis of Robot Foundation Model

Deep learning has demonstrated superior performance and generalization capabilities compared to traditional methods. Despite this, its nature as a "black box" algorithm often raises concerns regarding trustworthiness. The lack of transparency into the model's decision-making process makes it difficult to understand its actions. This section explores applications of our reasoning injection module, designed to improve model interpretability by illuminating its internal reasoning and explaining why certain actions might fail.

**How does the model generalize to new objects?** We conduct experiments on sorting tasks. Specifically, we ask the model to identify and sort unseen objects, that are not in the train data. We place four previously unseen objects on the plate, a stuffed toy cat, a pair of green gloves, a dark toy car, and a screwdriver.

From this experiment, we observe several interesting findings. First, the model demonstrates the ability to analogize to new objects. For instance, while the model has not been trained on a screwdriver, it correctly categorizes it as a hex key due to their visual similarity. This indicates that the model can generalize to new objects, not by direct recognition, but by comparing the object's semantic features with those of known objects. Second, the model can identify the color of some objects; for example, it categorizes the green glove as "green glove" and the stuffed toy cat as "brown toy cat." Although these experiments are not exhaustive, they represent an important step towards the explainable generalization of robot models.

**Failure case analysis via self-generated reasoning.** A key advantage of our model is its ability to generate natural language rationales alongside its output actions, providing valuable insight into its decision-making process. By observing these reasoning phrases, we can understand what the model is "thinking" at each step. For instance, as

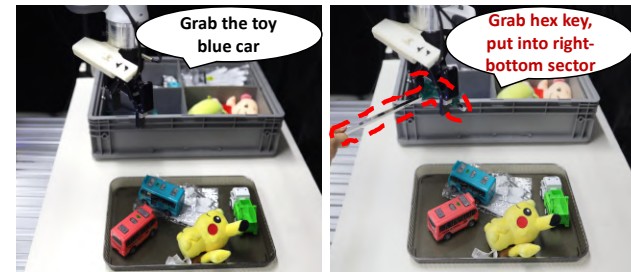

Figure 6: **What internal processes guide a model's actions?** We illustrate this using an example of DiVLA's reasoning, inferred from shifts in its behavior based on changes to the target object. The model initially intends to grasp a "toy blue car", but when presented with a hex key, its reasoning instantly redirects to sorting the "hex key" instead. This demonstrates one approach to interpreting a robot's actions.

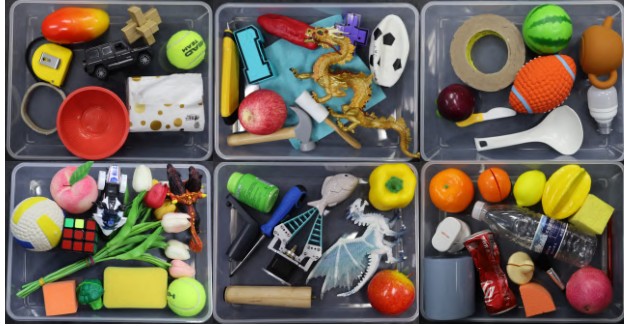

Figure 7: Some of the *unseen objects* used for evaluation in the zero-shot bin-picking tasks.

illustrated in Figure 6, the model might initially identify a toy car and generate the reasoning, "grabbing the toy car," indicating its intent to pick it up. However, if we intervene by placing a hex key in the gripper instead, the reasoning dynamically shifts to "grabbing the hex key." This adaptive reasoning allows the model to adjust its subsequent actions, correctly sorting the hex key despite the unexpected change. This dynamic reasoning not only enhances the transparency and interpretability of the model's actions but also contributes to its robustness. The integrated reasoning module allows for a form of self-correction, where the generated rationale guides and refines the action output, leading to more reliable performance even in the face of unexpected situations or perturbations. This capability suggests that the model is not simply executing pre-programmed actions, but rather engaging in a more flexible, context-aware decision-making process, akin to an internal dialogue.

## 4.5. Zero-Shot Bin Picking of Unseen Objects

This section evaluates instance generalization for DiVLA, focusing specifically on the bin-picking task—a standard benchmark for assessing robot model performance. In our

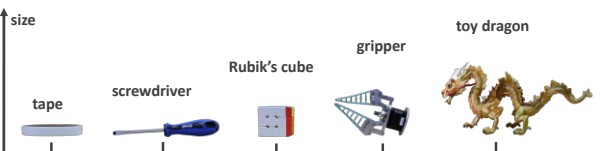

Figure 8: Examples of various unseen objects in zero-shot bin-picking tasks. The unseen objects vary significantly in size.

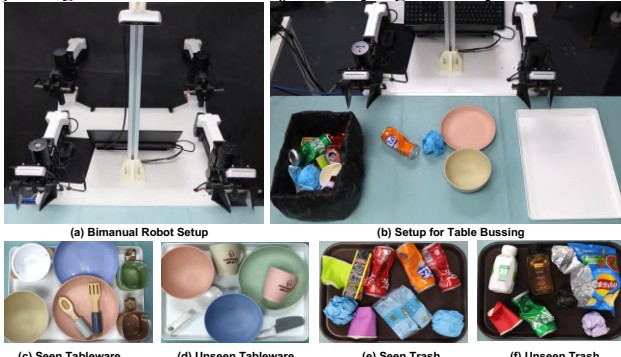

Figure 9: (a) Environmental setup for the bimanual robot, (b) Table bussing setup, (c-f) All tableware and trash items used in the bussing table task evaluation.

evaluation, we use *102* unique objects, ***none of which were included in the training data.*** Some of these objects are shown in Figure 7. We modify the task instruction to "move any object on the right panel to the left basket." Figure 2 (right) illustrates the experimental setup. We show some of the objects used in this test in Figure 7. The challenge in this evaluation lies in the significant variability across objects, which includes not only differences in dimensions but also distinct color patterns, textures, and degrees of deformability. Such variability aims to emulate real-world scenarios where robots must adapt to diverse and unpredictable items. Figure 8 provides examples of five differently sized objects from this experiment.

The experimental results are depicted in Figure 4. DiVLA achieves a success rate of 63.7%. In comparison, the success rates of the Diffusion Policy, Octo, TinyVLA, and OpenVLA are 8.9%, 19.6%, 23.5%, and 28.4%, respectively. These results indicate that DiVLA is resilient across a broad range of object shapes and sizes, other models often fail due to reliance on object-specific features that may not generalize well to new instances. It highlights the potential for applications in dynamic, unstructured environments where robots encounter unfamiliar objects and must perform tasks with minimal human intervention.

### 4.6. Adapt to Real-World Bimanual Robot

In this section, we examine DiVLA's adaptability to a dual-arm robot. Inspired by $\pi_0$ (Black et al., 2024), we designed a table bussing task that involves clearing table with various

Table 2: **Experimental results for Table Bussing on real bimanual robot.** Our method significantly outperforms both Diffusion Policy and OpenVLA by a large margin.

| Scenarios | Diffusion Policy | OpenVLA | **DiVLA-2B** |
|---|---|---|---|
| Seen | 45.8 | 0 | **72.9** |
| Mixed | 31.2 | 0 | **70.8** |

objects. This task was adapted for a bimanual robot setup: all tableware should be placed on a panel to the left, while trash items are dropped into a bin on the right. Similar to our factory sorting task, we evaluated the model's performance using both seen objects and a combination of seen and unseen objects. The environment setup, along with all objects used for training and evaluation, is shown in Figure 9. Our evaluation consisted of twelve trials, with three to five objects randomly placed on the table. The success rate is computed by how many objects are correctly placed.

Our results show that our model successfully completes tasks in most cases when the object has appeared in the training data, achieving an average success rate of 72.9% on seen objects. In contrast, the Diffusion Policy and OpenVLA achieve 45.8% and 0% success rates. For tasks involving both seen and unseen objects, DiVLA achieves up to a 70.8% success rate, a slight decrease in success rate compared to the seen one, showing remarkable generalization to objects with different colors and shapes. Finally, DiVLA demonstrates the ability to recognize unseen objects, particularly by responding to object color. For example, it categorizes a Sprite can as a "green can" and correctly places it in the trash bin. This observation further supports the idea that reasoning contributes to generalization.

### 5. Conclusion

In this work, we present DiVLA, a state-of-the-art vision-language-action model that delivers strong performance in both simulations and real-world scenarios, including single-arm and dual-arm robots. The core of our method lies in combining the next-token prediction objectives and diffusion models: the former for task reasoning and the latter for action prediction. We introduce a reasoning reuse module to enhance action generation. Through extensive evaluations in both simulations and across multiple real-world embodiments, we demonstrate that DiVLA outperforms several SOTA robot models. Additionally, we show that DiVLA has robust generalization capabilities, adapting effectively to new instructions, tasks, and environments. Our research offers a novel perspective on designing VLA models, encouraging a rethinking of how reasoning can be reused to facilitate end-to-end policy learning.

## Acknowledgements

This work is supported by the Sci-Tech Innovation Initiative by the Science and Technology Commission of Shanghai Municipality (24ZR1419000), the National Science Foundation of China (12471501), and the Science and Technology Innovation Action Plan of Shanghai under Grant 22511105400.

## Impact Statement

This paper presents work whose goal is to advance the field of Machine Learning. There are many potential societal consequences of our work, none which we feel must be specifically highlighted here.

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

# Supplementary Material

# 6. Evaluation Tasks and Detailed Results

## 6.1. Evaluation Tasks

As described in Section 4, we evaluate our DiVLA and baselines on multi-task learning, factory sorting, zero-shot bin-picking, and table bussing. In this section, we provide details on these tasks. All of these tasks are depicted in Figure 10 and 11.

For multi-task learning, we evaluate all methods on 5 tasks for multi-task learning and 5 visual generalization tasks. We evaluate each method with a total of 77 trials for multi-task learning and 45 trials for visual generalization. We provide the number of demonstrations and the average length of these tasks in 3. A brief description of these tasks is as follows:

- **Object selection.** In this task, we evaluate the model's ability to comprehend and act on user intent effectively. To do so, we design three distinct instructions, each requiring the robot to identify and pick up the appropriate object based on the user's expressed intent. These instructions are intentionally diverse to test the model's versatility and robustness in understanding varying contexts and nuances of user commands.

- **Flip the vertically placed pot.** In this task, we evaluate the model's commonsense reasoning abilities. Specifically, the model must determine whether a vertically placed pot is oriented to the right or left and accurately identify its direction to position it correctly. This test assesses the model's capacity to understand spatial orientation and apply logical reasoning to achieve the desired outcome.

- **Place cube inside box with closed lid.** In this task, we evaluate the model's ability to reason about and execute a multi-step operation that involves manipulating objects with physical constraints. The robot must first identify the box with the closed lid, open the lid, and then place the cube inside the box. This task assesses the model's planning, understanding of spatial relationships, and ability to perform actions in a sequence to achieve a desired goal.

- **Place cup onto the plate.** In this task, we test the model's ability to perceive and reason about the spatial arrangement of objects in a structured environment. A plate is placed on a two-tiered shelf, and the robot must first identify which tier the plate is located on. And the robot needs to carefully place the cup onto the plate without disturbing the surrounding setup. This task evaluates the model's spatial reasoning and precision in action.

- **Place cube into yellow/blue Box.** In this task, we assess the model's ability to follow instructions and demonstrate spatial reasoning. The robot is presented with an instruction specifying a designated box, either yellow or blue, and is required to accurately identify the correct box based on the given instruction. Once identified, the robot must then place a cube inside the chosen box. This task tests not only the model's comprehension of language-based directives but also its ability to integrate spatial awareness and execute precise actions.

**Setup for visual generalization.** In this setting, we assess the model's robustness and ability to generalize its visual perception to different environmental conditions. The robot is required to perform manipulation tasks under challenging visual variations, including randomly placed distractors, distracting lighting conditions, and a colorful background. These variations test the model's capacity to maintain focus on the primary task while filtering out irrelevant visual noise and adapting to dynamic visual environments. The goal is to ensure the robot can accurately identify and interact with objects despite significant deviations from standard settings.

Moreover, detailed descriptions for factory sorting, zero-shot bin-picking, and table bussing are provided in Sections 4.3, 4.5, and 4.6, respectively.

Table 3: Summarization for the number of demonstrations and average trajectory length for our real-world tasks.

| # | Task | # of Demonstrations | Avg. Traj. Length |
|---|---|---|---|
| 1 | Object selection | 160 | 91 |
| 2 | Flip the vertically placed pot | 120 | 137 |
| 3 | Place cube inside box with closed lid | 50 | 146 |
| 4 | Place cup onto the plate | 100 | 107 |
| 5 | Place cube into yellow/blue Box | 100 | 90 |

## 6.2. Detailed Results

In this section, we present the complete evaluation results for both the Franka and Bimanual AgileX robots, as detailed in Table 4. DiVLA-2B consistently demonstrates superior performance across the majority of tasks. Additionally, our findings highlight DiVLA-2B's robust visual generalization capabilities, effectively adapting to changes in the surrounding environment.

# 7. Implementation Details

For the baselines, we follow a uniform training strategy to ensure consistency. For OpenVLA, the original implementation uses only a single camera view. Since our method use three camera views on both single arm Franka robot and bimanual AgileX robot, it would be unfair to compare the vanilla implementation of OpenVLA that uses only a single camera view. Therefore, for the purpose of fair com-

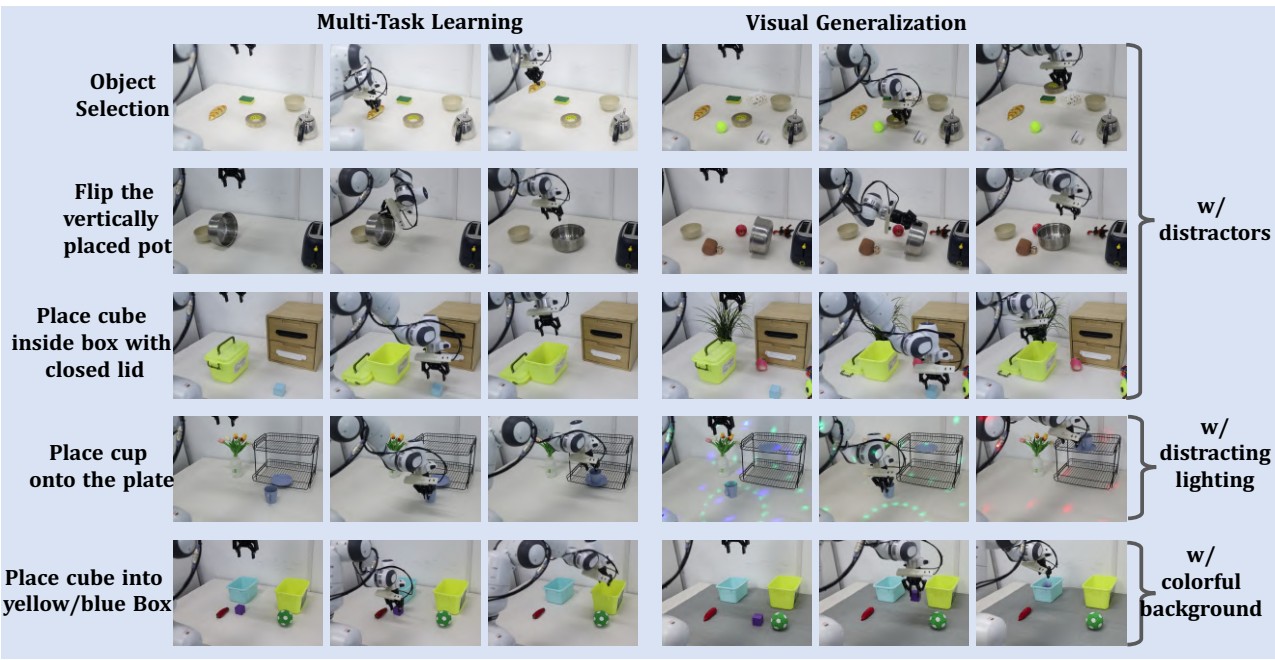

Figure 10: **Multi-task Learning and Visual Generalization.** We evaluate each method on multi-task learning and visual generalizations, including adding additional distractors, changing the background, and implementing colorful lighting. Each set of three images represents the initial state, intermediate state, and final state of one trial.

parison, we extend OpenVLA to utilize three camera views, processing each view through the same visual encoder and concatenating their outputs for further analysis. OpenVLA's pre-trained weights are used, and we fine-tune the model on our dataset with a learning rate of 2e-5. Training is conducted over 20 epochs, as we find that OpenVLA typically requires longer training times for convergence. For the Diffusion Policy, we employ DistilBERT to encode language instructions, adopting a methodology similar to YAY (Shi et al., 2024). Table 7 shows the performance of the single-view OpenVLA baseline. In the sorting task, the success rate drops significantly from 45.3% with three views to just 12.7% with a single view, highlighting the importance of our proposed multi-view approach.

## 8. More Experiments

### 8.1. View Shifting Generalization

Imitation learning often struggles to generalize its capabilities to novel viewpoints. In this study, we evaluate the view generalization performance of DP, OpenVLA, and DiVLA-2B in a factory sorting task. The experimental setup is illustrated in Figure 12. As shown in Table 6, OpenVLA exhibits poor generalization to view shifts. In contrast, DiVLA-2B demonstrates notable robustness in view generalization, achieving a success rate of 60%. This result highlights the advantages of leveraging pretraining on large-scale robotic datasets to enhance generalization capabilities.

### 8.2. Efficient Inference

Fast inference speed is critical for deploying VLA models in real-world applications. However, as models grow in size and complexity, their inference speed tends to decrease significantly on server-grade hardware. To address this challenge and enhance model performance, we deploy our models on the vLLM framework (Kwon et al., 2023). This approach leverages optimized infrastructure to maximize inference efficiency. In Table 5, we present a comparative analysis of control frequencies for DiVLA-2B, DiVLA-7B, and OpenVLA (a 7B model). Our results demonstrate remarkable speed improvements: the DiVLA-2B model achieves an impressive 82 Hz on an A6000 GPU, showcasing exceptional performance. Similarly, DiVLA-7B achieves a control frequency of 42 Hz, which is 8 times faster than OpenVLA at the same model size. These findings underscore the effectiveness of our optimizations in maintaining scalability without sacrificing speed, paving the way for broader real-world applicability of VLA models.

While vLLM accelerates our method, the improvement in inference time is less dramatic than reported for LLMs. Even without vLLM, DiVLA-2B and DiVLA-7B achieve respectable frame rates of 74Hz and 30Hz, respectively. Notably, this is still six times faster than OpenVLA, which has a similar model size (7 billion parameters).

Another key observation is that our model experiences significant performance degradation when evaluated at lower

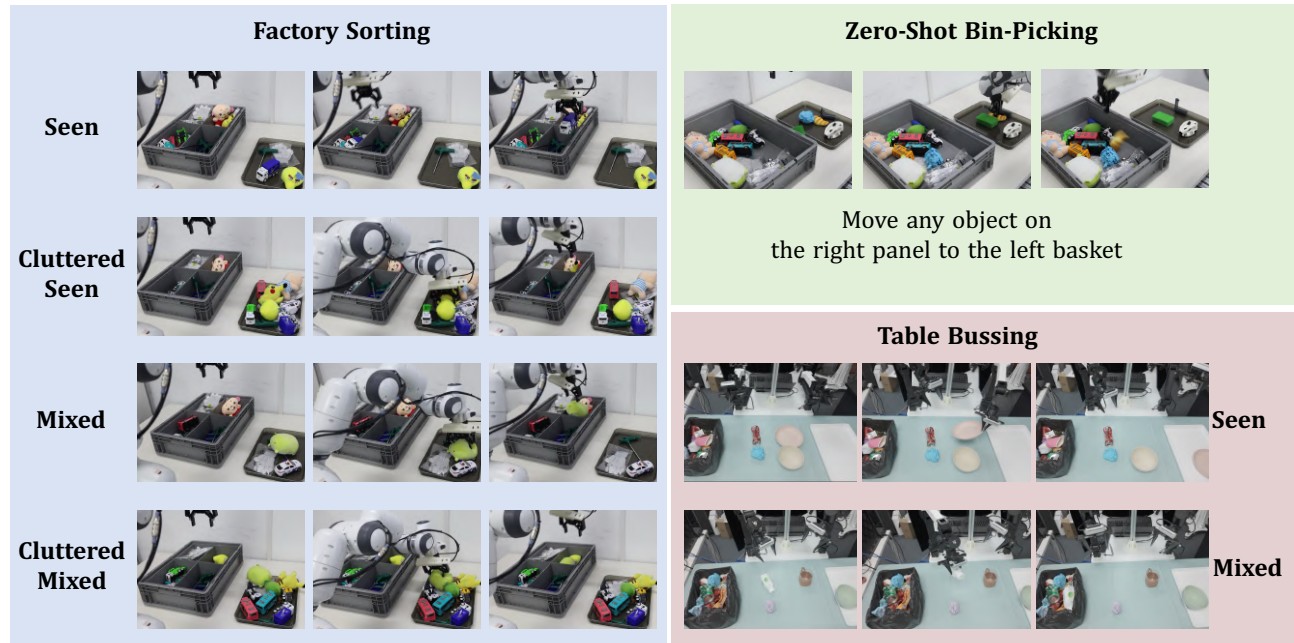

Figure 11: **Factory Sorting, Zero-Shot Bin-Picking and Table Bussing.** We further evaluate all robot policies on additional challenging tasks, including factory sorting, zero-shot bin-picking, and bimanual table bussing. These tasks involve previously unseen objects with diverse textures, varying heights, and different degrees of deformability (as illustrated in Figure 7 and Figure 8). Each set of three images represents the progression of a single trial, showcasing the initial state, intermediate state, and final state.

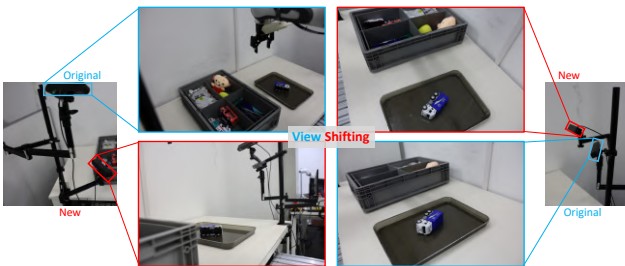

Figure 12: **View Generalization**. We evaluate DP, OpenVLA, and DiVLA-2B in the view shifting setting, where we use completely different camera positions to capture images. The blue part indicates the original camera positions, while the red part indicates the new camera positions.

bit precisions, such as 8-bit and 4-bit. This aligns with findings in OpenVLA, which highlight performance fluctuations introduced by quantization. These results suggest that current VLA models may require specifically designed quantization methods to maintain performance while achieving fast inference speeds at low precision.

### 8.3. Following Novel Instruction

The previous section primarily evaluated the model's performance on in-distribution task commands. In this section, we assess the model's ability to follow novel instructions, specifically focusing on its generalization to previously unseen commands. We introduce new instructions that prompt the model to pick up unseen items and follow sequential instructions. Specifically, the instruction templates are "Pick up {obj}." and "Pick up {obj} first, then pick up the {obj}, finally pick up {obj}." We tested the model on four objects: 1) a watermelon, 2) a lemonade, 3) a blue paper trash, and 4) a red pepper. This is an extremely challenging task, as these novel instructions are absent from both the Droid dataset and our collected data. We evaluated four new instructions, with results summarized in Table 9.

Our findings indicate that both OpenVLA and DiVLA-2B can recognize these unseen objects and perform basic pick-and-place tasks. However, when it comes to complex sequential tasks, OpenVLA fails to interpret the instructions accurately, instead randomly selecting items. In contrast, our method correctly follows the instructions, picking up objects in the specified sequence. We hypothesize that by learning to decompose long tasks into subtasks, our method acquires a generalized ability to understand complex, multi-step instructions. While OpenVLA can execute simpler commands like "Pick up the watermelon," it struggles with more advanced instructions requiring item selection in a specific order. Additionally, we observe a decrease in grasping precision when the model processes novel instructions, indicating that the novelty of instructions introduces further complexity to task execution.

Table 4: **Detailed Experimental Results on both Franka and AgileX Aloha.**

| Type | Task | Trails | DP | TinyVLA | Octo | OpenVLA | DiVLA-2B |
|------|------|--------|-----|---------|------|---------|----------|
| Multi-Task Learning | Object Selection | 33 | 22 | 23 | 19 | 24 | **33** |
| | Flip the Vertically Placed Pot. | 11 | 4 | 5 | 3 | 2 | **11** |
| | Place Cube into Yellow/Blue Box | 11 | 0 | 3 | 3 | 6 | **10** |
| | Place Cup onto Plate | 11 | 4 | 5 | 1 | 4 | **7** |
| | Place Cube inside Box with Closed Lip | 11 | 0 | 4 | 2 | 2 | **7** |
| Visual Generalization | Object Selection | 9 | 1 | 4 | 4 | **5** | 4 |
| | Flip the Vertically Placed Pot. | 9 | 1 | 4 | 1 | 1 | **6** |
| | Place Cube into Yellow/Blue Box | 9 | 0 | 2 | 0 | 3 | **6** |
| | Place Cup onto Plate | 9 | 2 | 2 | 0 | 3 | **6** |
| | Place Cube inside Box with Closed Lip | 9 | 0 | 1 | 2 | 0 | **4** |
| More Challenging Tasks | Factory Sorting (Seen) | 80 | 27 | 39 | 33 | 45 | **61** |
| | Factory Sorting (Mixed) | 80 | 11 | 25 | 27 | 36 | **50** |
| | Factory Sorting (Cluttered Seen) | 65 | 16 | 21 | 17 | 30 | **43** |
| | Factory Sorting (Cluttered Mixed) | 65 | 6 | 14 | 15 | 22 | **40** |
| | Zero-Shot Bin-Picking | 102 | 10 | 24 | 4 | 29 | **65** |
| | Table Bussing (Seen) | 48 | 22 | - | - | 0 | **35** |
| | Table Bussing (Mixed) | 48 | 15 | - | - | 0 | **34** |

Table 5: **Inference speed for DiVLA.** We report the inference speed for DiVLA on A6000 GPU.

| Method | DiVLA-2B | DiVLA-7B | OpenVLA-7B |
|--------|----------|----------|------------|
| Control Frequency | 82Hz | 42Hz | 5Hz |

Table 6: **Experimental results for view shifting generalization.** We conduct view shifting generalization on factory sorting task. The experimental setup is shown in Figure 12. We report the success rate for each policy. Each method is evaluate with 5 trails.

| Task \ Method | DP | OpenVLA | DiVLA-2B |
|---------------|-----|---------|----------|
| Factory Sorting | 0 | 0 | 60 |

Table 7: **Ablation study on OpenVLA using one camera view and three camera views.** For a fair comparison, our main experiments evaluate OpenVLA using three camera views with existing methods. This section presents an ablation study comparing these results with those obtained using a single-camera view to demonstrate the benefits of our multi-view approach.

| Method/Sorting | Seen | Mixed | Cluttered Seen | Cluttered Mixed | Avg. |
|----------------|------|-------|----------------|-----------------|------|
| OpenVLA (3 views) | 56.2 | 46.2 | 45.0 | 33.8 | 45.3 |
| OpenVLA (1 view) | 26.4 | 17.5 | 4.8 | 2.1 | 12.7 |

Table 8: **Ablation study on reasoning injection module.**

| Model \ Tasks | In-Distribution | | | | | |
|---------------|--------|--------|--------|--------|--------|------|
| | Task 1 | Task 2 | Task 3 | Task 4 | Task 5 | Avg. |
| DiVLA-2B | **100** | **100** | **63.6** | **63.6** | **90.9** | **83.6** |
| w/o reasoning injection | 66.7 | 66.7 | 45.5 | 45.5 | 27.3 | 50.3 |

## 8.4. Ablation Study on Reasoning Injection Module

Our core contribution is the proposed reasoning module, which enables the network to complete long-horizon tasks via direct prompting. This section presents an ablation study of this module, following the real-world experimental setup described in Section 4.2. We train a model using the same methodology, but without the reasoning injection module. Table 8 presents the results. We observe a significant performance drop compared to our baseline model when the reasoning injection module is removed. We hypothesize that the reasoning module facilitates task decomposition, resulting in more fine-grained action generation, which simplifies the learning process.

## 8.5. Model Scaling

An important consideration for pure end-to-end neural networks is assessing their ability to scale effectively. As the model size and data volume grow, the performance of the model should ideally improve. To evaluate this scalability, we conducted additional experiments on two challenging tasks: factory sorting and zero-shot bin-picking. These tasks provide rigorous benchmarks to measure how increased model capacity and pre-training data influence performance. The detailed experimental results are presented in Table 10.

Our analysis highlights significant performance improvements when larger models and more extensive pre-training datasets are utilized. For example, the DiVLA-7B model demonstrates substantial gains over the DiVLA-2B model, achieving improvements of 8.7% and 3.0% in success rates for the factory sorting and bin-picking tasks, respectively. This underscores the benefits of scaling model size, which enables better representation learning and decision-making capabilities.

Table 9: **Experimental Results for Following Novel Instruction.** We report the success rate, calculated as the number of successful trials divided by the total trials, for the novel instruction-following abilities of OpenVLA and DiVLA-2B. Our method successfully picks up items based on given instructions and can follow sequential instructions to retrieve items in the correct order.

| Model \ Tasks | Following Novel Instruction on Four Tasks | |
|---|---|---|
| | Watermelon | Watermelon → Blue Paper Trash → Lemonade |
| OpenVLA (Kim et al.) | **2/3** | 0/3 |
| **DiVLA-2B** | 2/3 | **2/3** |
| Model \ Tasks | Red Pepper → Blue Cup | Watermelon → Lemonade → Blue Paper Trash |
| OpenVLA (Kim et al.) | 0/3 | 0/3 |
| **DiVLA-2B** | **1/3** | **1/3** |

Moreover, scaling up to 72B parameters yields even greater performance boosts. These larger models not only achieve higher success rates in in-distribution scenarios but also exhibit superior generalization to out-of-distribution setups. This improvement reflects the models' enhanced ability to handle variability in task environments, unseen object properties, and dynamic conditions.

Table 10: **Experimental results for model scaling.**

| Tasks/Models | DiVLA-2B | DiVLA-7B | DiVLA-72B |
|---|---|---|---|
| Sorting | 66.2 | 74.9 | **82.4** |
| Bin Picking | 63.7 | 66.7 | **75.9** |

## 8.6. DiVLA can do Visual-Question-Answering

Previous works, such as RT-2 (Brohan et al., 2023) and ECoT (Zawalski et al., 2024), have suggested that co-training with vision-language data helps preserve basic conversational functionality for Vision-Language Action (VLA) data. In this section, we demonstrate that despite not being co-trained with vision-language data, our method retains its chatting capability and is proficient in performing common visual-question-answering (VQA) tasks. However, these works have neither provided concrete examples of chat interactions nor open-sourced their results, leaving it unclear to what extent conversational capabilities are retained in such models. We give a few examples in Table 11.

When prompted to identify objects, we observe that the model demonstrates the ability to recognize some items accurately, such as tulips and oranges. However, it struggles with more nuanced distinctions; for instance, DiVLA mistakenly identifies a toy dragon as a toy tiger, likely due to its reliance on color features rather than other distinguishing characteristics like shape or texture.

Interestingly, we observe that the model demonstrates strong sensitivity to color, consistently providing correct answers to all three questions focusing on an object's color. This highlights a notable strength in color recognition but also

Table 11: **VQA for DiVLA.** We test DiVLA's ability to answer questions based on visual signals.

| Question | Object/Scene | Answer | Correct |
|---|---|---|---|
| What is the object? | | Toy Tiger | ✗ |
| | | Tulip | ✓ |
| | | Orange | ✓ |
| What is the object's color? | | Brown | ✓ |
| | | Blue | ✓ |
| | | Green | ✓ |
| Describe the scene. | | The cube is on the right side of yellow pepper | ✓ |
| | | The ball is on the top of a holder | ✗ |

reveals limitations in processing broader visual features for complex tasks. To further assess its capabilities, we tested the model with a simple scene description task, requiring it to describe spatial relationships between objects. The model successfully interpreted spatial relationships, such as identifying that one object is on the right or on top of another. However, it often failed to recognize objects correctly. For example, DiVLA misclassified a football as a regular ball.

Notably, these objects were not included in the model's pre-training or fine-tuning data. However, the pre-trained VLM may have encountered similar objects during its pre-training phase, enabling it to recognize them correctly in some cases. This observation underscores the importance of leveraging pre-trained VLMs as a foundation for end-to-end visuomotor learning, as they provide a strong prior understanding of visual concepts that can significantly enhance downstream performance in complex tasks.

