# OpenReview forum: "DiffusionVLA: Scaling Robot Foundation Models via Unified Diffusion and Autoregression"
_ICML.cc/2025/Conference — ICML 2025 poster_

### Official Review · Reviewer_bZhD · 2025-03-11

**Overall Recommendation:** 2

**Summary:**

This paper proposes a novel VLA framework that integrates NTP with a diffusion process. The writing is logical, and the experimental evaluation is thorough. While the core idea is intriguing and somewhat similar to $\pi_0 $[1], the latter was released in October, close to the ICML submission deadline in January, which may explain the absence of a direct comparison. However, it would still strengthen the paper if the authors highlighted the specific differences between their approach and $\pi_0 $[1],

[1] Black, Kevin, et al. "$\pi_0 $: A Vision-Language-Action Flow Model for General Robot Control." arXiv preprint arXiv:2410.24164 (2024).

**Claims And Evidence:**

yes

**Essential References Not Discussed:**

Black, Kevin, et al. "$\pi_0 $: A Vision-Language-Action Flow Model for General Robot Control." arXiv preprint arXiv:2410.24164 (2024).

**Experimental Designs Or Analyses:**

yes

**Methods And Evaluation Criteria:**

yes

**Other Comments Or Suggestions:**

- Inference speed: After reviewing the supplementary materials, it appears that the high inference speed primarily arises from the use of the vLLM framework rather than the design of the method itself. To enhance clarity, it would be beneficial to explain in the main paper how vLLM is used to speed up inference and provide more details about its underlying implementation.

- Method details:
  1. The paper introduces a new VLA architecture but does not include a detailed architectural diagram. Specifically, the "Projection layer" and "action decoder" mentioned starting at Line 212 are absent from Figure 1, making it difficult to understand their functions and how they interact with other components. Including a comprehensive architecture figure that clearly illustrates these modules and their connections would greatly improve clarity.
  2. In Figure 1, it appears that the reasoning tokens are transformed into language output. However, the method section does not mention any language output. Additionally, it remains unclear whether the ground truth (GT) for $L_{NTP }$ is based on action supervision or language supervision.

- Others:
  1. The paper references Table 1 on page 6, but the table is actually located on page 4, which disrupts the flow of the experimental discussion. A similar issue occurs with Figure 3. Please ensure that the references match their corresponding figures/tables to improve readability.
  2. Although the model can perform corrections, it remains unclear why certain failures still occur. The authors should include a failure case analysis section to clarify the limitations of the model and suggest potential improvements.

**Other Strengths And Weaknesses:**

Refer to Other Comments Or Suggestions

**Questions For Authors:**

Refer to Other Comments Or Suggestions

**Relation To Broader Scientific Literature:**

Related

**Theoretical Claims:**

yes

---

> ### Author Rebuttal · Authors · 2025-04-01
>
> Thank you for your thorough review and valuable feedback. We have addressed each point below. Please see the following responses for details.
>
> ## 1. The inference speed of DiVLA(Same as R3#1)
>
> This issue was partially addressed in our response to R13rJ#1. Bellow, we fristly clarify why DiVLA-7B has similar number of parameters but is 8 times faster than OpenVLA-7B, followed by an explanation of how vLLM optimizes inference efficiency in VLMs.
>
> 1) **Inference Speed.** Diffusion-based VLA operates significantly faster than autoregressive VLA in robot action generation. In robotics, inference speed refers to the number of actions produced per second. For instance, using the Franka Emika with 7 degrees of freedom, OpenVLA must predict 7 tokens for each action. In a typical setup, the model needs to predict 30–60 actions for each incoming observational state, requiring 210–420 tokens for prediction. In contrast, diffusion-based VLA, such as DiVLA, only needs 10 denoising steps to predict 30–60 actions. This conclusion is supported by FAST [1], which finds that diffusion-based VLA is much faster than autoregressive VLA. In summary, DiVLA benefits from significantly faster inference times due to its use of a diffusion process, confirming the accuracy of our inference tests.
>
> 2) **DiVLA with vLLM.** Regarding our DiVLA with vLLM, the inference speed of VLM can achieve approximately 2× speedup. vLLM enhances VLM inference efficiency through PagedAttention, which prevents memory fragmentation by storing KV cache in fixed-size pages, reducing memory overhead and improving stability for long-sequence generation. Additionally, optimized CUDA kernels accelerate attention computation, resulting in significantly faster inference compared to traditional frameworks, often achieving 2–3× speedup.
>
> | Model | T_ntp \ T_diff (s) | Chunk Size |  Inference speed |
> | --- | --- | --- | --- |
> | OpenVLA | 0.27 \ - | 1 | 4HZ |
> | DiVLA w/o vLLM | 0.5 \ 0.06 | 16 | 29HZ |
> | DiVLA w/ vLLM | 0.3 \ 0.07 | 16 | 43HZ |
>
> ## 2. Details architectural diagram for DiVLA
>
> We apologize for the absence of the detailed architectural diagram. As you correctly pointed out, the “projection layer" and "action decoder" are crucial components of our framework but are not represented in Figure 1. To address this, we will include a more comprehensive diagram that clearly illustrates the entire architecture, including these specific modules and their interactions.
>
> The projection layer consists of two MLP layers with LayerNorm. It bridges the VLM's embedding to the diffusion model, aligning their dimensions. Additionally, our action decoder follows the standard diffusion policy design [2]. It takes the embeddings from the VLM as conditions and predicts the noise based on the noisy actions, repeating this process *k* times to obtain the final action chunk.
>
> ## 3. Clarify on Language Generation and  the L_NTP loss function
>
> In Section 3.2 Model Design Choices, we state that our training loss comprises tow part: the diffusion loss L_diff and the NTP loss L_ntp. The ground truth for L_ntp is based on language supervision. Consequently, we can generate reasoning tokens (i.e. language output) during inference.
>
> ## 4. Failure case analysis
>
> We have failure case analysis (**page 7 lines 378-384**) which systematically examines how dynamic reasoning enables action self-correction. To extend this discussion, we propose a **dual-aspect failure taxonomy**: **erroneous reasoning chains** and **action execution errors**. As demonstrated in Table through controlled experiments on the bin-picking task, we evaluate 4 trials per object across both seen and unseen categories. Seen objects are appeared in robot data while unseen objects are not. Our model could successfully pick up those objects when facing seen objects and we empirically observe that action always fails when the model **cannot recognize the unseen objects** though. We attribute the recognition degradation to robot-data overfitting as exclusive training on robotic demonstrations diminishes the base VLM's open-vocabulary recognition capacity. Thus a potential solution is co-training the VLA with robot data and vision-text data, preserving both generalizable visual concepts and action-specific skills.
>
> | success/object | Watermelon(unseen) | dragon(unseen) | Orange(unseen) | toy car(seen) | hex key(seen) |
> | --- | --- | --- | --- | --- | --- |
> | Reasoning | 50 | 0 | 75 | 100 | 100 |
> | Action | 50 | 0 | 50 | 100 | 100 |
>
> ## 5. Typo problem
>
> Thank you for your careful review. We appreciate your feedback regarding the misreferenced table and figure. These issues have been corrected in the next version of the manuscript, which will be updated accordingly.
>
> [1]. FAST: Efficient Action Tokenization for Vision-Language-Action Models
>
> [2]. Diffusion policy: Visuomotor policy learning via action diffusion.

---

### Official Review · Reviewer_13rJ · 2025-03-12

**Overall Recommendation:** 2

**Summary:**

DiVLA is a VLA model that connects a VLM with a diffusion model to enable both reasoning and action generation in robotics. It builds upon a pre-trained Vision-Language Model (VLM) for text-based reasoning while incorporating a diffusion model to learn robotic actions through a noise-denoising process. DiVLA introduces a reasoning injection module, which embeds reasoning outputs directly into the policy head, enhancing decision-making and interpretability. This framework demonstrates strong generalization and robustness in basic tasks (e.g., pick and place) compared to previous methods.

**Claims And Evidence:**

yes

**Essential References Not Discussed:**

This paper provides a thorough discussion of related works.

**Experimental Designs Or Analyses:**

Please refer to 'Other Strengths and Weaknesses.'

**Methods And Evaluation Criteria:**

Please refer to 'Other Strengths and Weaknesses.'

**Other Comments Or Suggestions:**

It is recommended that the authors properly retest the model's inference speed. If DiVLA generates language reasoning during inference using an autoregressive next-token prediction approach, then DiVLA-7B's inference speed should be comparable to OpenVLA.
For instance, the process of DiVLA-7B generating 7 words in its LLM follows a similar inference logic to OpenVLA generating 7 action bins. However, if DiVLA does not generate language reasoning during inference, then how are Reasoning Tokens constructed?

**Other Strengths And Weaknesses:**

Strengths:

1. Attaching the diffusion module after the VLM to construct a VLA model is the mainstream action prediction approach, similar to TinyVLA and Pi_0.

2. Unlike previous diffusion-based methods, DiVLA further incorporates Reasoning Tokens to enhance action generation, representing an innovative exploration.

3. DiVLA exhibits strong zero-shot bin-picking capabilities.

Weaknesses:

Q1 My primary concern is the inference speed.
1.1 The authors claim that DiVLA-7B's inference speed is 8 times faster than OpenVLA at the same model size, which appears unrealistic. OpenVLA performs next-token prediction for 7 action tokens (discrete end-effector poses), whereas DiVLA additionally requires language next-token prediction (e.g., "Pick up the Rubik’s cube"). Given this, the inference speeds of both models should be comparable. The authors should retest and verify DiVLA's speed.
1.2 If LLM acceleration tools such as vLLM were utilized, this should be explicitly stated in the main text.

Q2 Are Reasoning Tokens and Action Tokens both used as conditions for the Diffusion Model? If so, while Reasoning Tokens are obtained through next-token prediction, how are Action Tokens generated? Are they directly derived from the question token output of the VLM?

Q3 DiVLA-2B and DiVLA-7B were pretrained on the Droid dataset, which is relatively limited compared to OXE. Could the model’s zero-shot bin-picking ability be attributed to its memory of the gray experiment tray (e.g., picking all objects within the tray) rather than genuine generalization to object semantics? For example, how would the model perform if objects were placed at different positions on the table without the tray?

**Questions For Authors:**

Please refer to 'Other Strengths and Weaknesses.'

**Relation To Broader Scientific Literature:**

Effectively constructing a VLA model for robotics.

**Theoretical Claims:**

Application paper without making theoretical claims.

---

> ### Author Rebuttal · Authors · 2025-04-01
>
> Thanks for your careful review and valuable comments. We address each question below.
>
> ## 1. Why DiVLA-7B has similar number of parameters but is 8 times faster than OpenVLA-7B?
>
> Thank you for pointing out this question.
>
> 1) Diffusion-based VLA operates significantly faster than autoregressive VLA in robot action generation. In robotics, inference speed refers to the number of actions produced per second. For instance, using the Franka Emika with 7 degrees of freedom, OpenVLA must predict 7 tokens for each action. In a typical setup, the model needs to predict 30–60 actions for each incoming observational state, requiring 210–420 tokens for prediction. In contrast, diffusion-based VLA, such as DiVLA, only needs 10 denoising steps to predict 30–60 actions. This conclusion is supported by FAST [1], which finds that diffusion-based VLA is much faster than autoregressive VLA. In summary, DiVLA benefits from significantly faster inference times due to its use of a diffusion process, confirming the accuracy of our inference tests.
>
> 2) To further ablate the impact of vLLM, we report the following test on an A6000 GPU, where T_ntp and T_diff represent the time spent on the NTP and diffusion processes, respectively.
>
> | Model | T_ntp \ T_diff (s) | Chunk Size |  Inference speed |
> | --- | --- | --- | --- |
> | OpenVLA | 0.27 \ - | 1 | 4HZ |
> | DiVLA w/o vLLM | 0.5 \ 0.06 | 16 | 29HZ |
> | DiVLA w/ vLLM | 0.3 \ 0.07 | 16 | 43HZ |
>
> ## 2. Clarify the role of reasoning and action tokens in DiVLA
>
> We appreciate you highlighting this.
>
> 1). **Are both tokens used as conditions for diffusion model?** Yes, both action and reasoning tokens serve as different conditions on action denoising. Specifically, action tokens mainly include visual embeddings providing global observations and raw instruction tokens. On the other hand, reasoning tokens encapsulate hierarchical substep-level features that decompose complex tasks into temporally executable steps.
>
> 2). **How action tokens are generated?** Action tokens refer to vanilla visual tokens and tokens from raw instruction processed by VLM backbone. They are not generated autoregressively.
>
> ## 3. Whether DiVLA's ability to perform zero-shot bin picking is attributed to its generalization capability or spatial memory?
>
> Thank you for your insightful question. To answer your question, the DiVLA’s ability to perform zero-shot bin-picking definitely stems from its strong generalization capability.
>
> As outlined in page 8 line 410-420, the bin picking experiment used 102 unique objects that were entirely absent from the training set, ensuring that the model had to generalize to novel objects.  In Figure 8, we shows that the objects have varying size and height. Object like white tape is only 1cm high, while the toy dragon is 10cm. Also, in Figure 7, we present some unseen objects that we used for zero-shot bin picking tasks. It can be observed that these objects exhibited substantial variations in shape, size, color, texture, and deformability, ensuring that the model could not rely on simple heuristics such as shape consistency or fixed spatial features. Our model, DiVLA, achieved a 63.7% success rate, substantially outperforming baselines (Diffusion Policy: 8.9%, Octo: 19.6%, TinyVLA: 23.5%, OpenVLA: 28.4%), underscoring its superior generalization for unseen objects.
>
> [1]. FAST: Efficient Action Tokenization for Vision-Language-Action Models

---

> > ### Comment · Reviewer_13rJ · 2025-04-04
> >
> > Thank you for the rebuttal. I intend to keep my rating (weak reject) and have a few questions to discuss with the authors.
> >
> > **1. Reasoning Tokens and Inference Speed**
> >
> > First, I understand that DDIM-based action denoising for the diffusion head (e.g., Pi₀, CogAct, TinyVLA) is generally faster than autoregressive action generation through LLMs (e.g., RT-2, OpenVLA). However, according to Figure 1 and the description in the Method section, if the DiVLA model performs language reasoning, it would still require autoregressive next-token prediction.
> > In other words, since action tokens already include all feedforward visual and language tokens, how are the reasoning tokens generated?
> > If the language reasoning outputs are not autoregressively generated and the corresponding reasoning tokens are not reused as conditions, then what fundamentally distinguishes DiVLA from Pi₀ or TinyVLA?
> > Therefore, I would appreciate a direct and detailed clarification from the authors on how the reasoning tokens are obtained.
> > If it is difficult to explain the generation process of reasoning tokens in words, could the authors kindly share the relevant code for clarification?
> >
> >
> >
> >
> > Second, if DiVLA requires generating reasoning tokens through autoregressive next-token prediction, this would significantly slow down the model’s inference speed. If the fast inference speed is due to the use of action chunking (where the control frequency equals the model inference speed multiplied by the action chunk size), this should be clearly stated in the paper. Additionally, it would be helpful to clarify whether the method employs temporal ensembling as in ACT, or instead directly executes a sequence of 16 actions before receiving a new image observation and language instruction. If temporal ensembling is used, the robot's control frequency would be reduced. If not, how is temporal consistency in the action outputs ensured?
> >
> >
> > **2. Generalization in Real-World Scenarios**
> >
> > Through both qualitative and quantitative evaluations, the paper demonstrates that DiVLA exhibits generalization capabilities across diverse object shapes, sizes, colors, textures, and deformability. However, could the authors further clarify whether in the unseen-object real-world task shown in Figure 7, the performance remains robust even when the object is placed directly on the table, rather than inside a tray?
> >
> >
> > **3. Lack of detailed information**
> >
> > Finally, I believe that incorporating language reasoning for substep-level or atomic tasks is a valuable contribution that sets this work apart from other diffusion-based VLA methods.
> > However, I hope the authors can further improve the paper to meet the standards of the ICML conference, as it currently lacks several important details. For instance, how is the substep-level language planning GT constructed in DiVLA? Is substep reasoning only performed at the beginning of each task, or does it accompany the entire task process? Are action chunking and temporal ensembling techniques used? Which subsets of the OXE dataset were used for pretraining, and how many trajectories were included? How many iterations and GPU hours were required for pretraining? Please feel free to correct me if I have misunderstood any part of the paper.

---

> > > ### Author Response · Authors · 2025-04-08
> > >
> > > Thanks for your careful review and valuable comments. We address each question below.
> > >
> > > **1.1  Direct and detailed clarification on how the reasoning tokens are obtained.**
> > >
> > > Thank you for raising this point. Reasoning tokens are generated autoregressively, same as standard LLMs/VLMs, and are then reused as conditioning inputs for the diffusion process via our novel reasoning injection module. For a more direct understanding, we have provided the source code: [https://anonymous.4open.science/r/divla_anonymous_icml-BFD4/README.md](https://anonymous.4open.science/r/divla_anonymous_icml-BFD4/README.md).
> > >
> > > **1.2 Why does DiVLA output reasoning (in language) but much faster than OpenVLA.**
> > >
> > > Our previous initial rebuttal have answered that because diffusion based method generated more action steps per second, therefore it is much faster than auto-regreesive based VLA method like OpenVLA.
> > >
> > > This is similar for reasoning generation. For instances, to predict 80 actions, OpenVLA generates 7 tokens per action, resulting in total 560 tokens. In contrast, DiVLA generates only 100 reasoning tokens, making it much faster than OpenVLA. Specifically, DiVLA predicts 16 future actions across 10 denoising steps, but executes only the first 8. To predict 80 executable actions, this demands 10 (80/8) diffusion processes and reasoning generations. Since the average length of reasoning tokens is 10, DiVLA's total token count is 100 (10×10), which is almost 1/5 of OpenVLA's count.
> > >
> > > Thus, DiVLA generates far fewer tokens than OpenVLA for the same length of actions and can process multiple actions concurrently, enables DiVLA to achieve significantly faster inference speeds.
> > >
> > > **1.3 Does DiVLA use temporal ensembling?**
> > >
> > > We appreciate your point. Our method does not employ temporal ensembling and utilizes only action chunking, a common practice in prior works such as Pi0, TinyVLA, DP3[1], DP[2].
> > >
> > > There might be a misunderstanding where responsiveness is being conflated with temporal consistency.  As clearly stated in DP[2], action chunking involves predicting a receding future action sequence, which inherently promotes temporal action consistency. Responsiveness refers to the speed at which the model reacts to observation updates which will be weakened by large size of action chunk.
> > >
> > > DP has conducted experiments exploring the trade-off between action horizon and responsiveness. To further optimize this balance, our method predicts future 16 actions but executes only the initial 8 at a 16Hz control frequency. This results in observation updates every 0.5 seconds, aligning with Pi0.
> > >
> > > **2 Does DiVLA remain robust when the object is placed on the table?**
> > >
> > > Thanks for pointing this out. Yes, DiVLA remains robust even when object is placed directly on the table. As demonstrated below, DiVLA achieves a 47.1% success rate, tripling the performance of OpenVLA. Furthermore, DiVLA exhibited greater robustness, with a performance decline of only 26.1% compared to OpenVLA's 44.7%. Those outcomes highlight DiVLA’s capability of handling complex generalization settings.
> > >
> > > | Models | placed in tray | placed on table | Relative Decrease ↓ (%) |
> > > | --- | --- | --- | --- |
> > > | DiVLA | 63.7 | 47.1 | 26.1 |
> > > | OpenVLA | 28.4 | 15.7 | 44.7 |
> > >
> > >
> > > **3.1 How is the GT substep reasoning constructed?**
> > >
> > > Thanks for pointing this out. We converted all data into video format and employed Gemini to annotate the robot's actions in the videos. To ensure consistency, we predefined multiple sets of substep templates for each task, allowing Gemini to randomly select the template for annotation.
> > >
> > > **3.2 How is substep reasoning performed?**
> > >
> > > Thanks for feedback. Our DiVLA performs substep reasoning throughout the entire task process. Notably, DiVLA generates one substep at a time based on the current observation rather than producing all substeps at once. This approach allows DiVLA to gain a clear understanding of the task's progress by evaluating its current state and determining the next appropriate substep.
> > >
> > > **3.3 Which subset of OXE is used and howmany trajectories are included?**
> > >
> > > Thank you for pointing this out. Since OXE includes a wide variety of embodiments across different settings, and some of the data is of low quality (e.g., super low resolution, 80x80), we filtered the data based on sufficient resolution, language annotations, single-arm configuration, and an appropriate task duration. As a result, approximately 9K trajectories are included.
> > >
> > > **3.4 Pretraining time (GPU hours).**
> > >
> > > Thank you for your feedback. We have had a comprehensive discussion on training efficiency in response to **Reviewer fKzV#1**, and you can refer to that section for details. For simplicity, DiVLA-2B was pretrained for only 2.5 epochs, equivalent to 155 H800 GPU hours.
> > >
> > > [1]. 3D Diffusion Policy: Generalizable Visuomotor Policy Learning via Simple 3D Representations RSS 2024
> > >
> > > [2]. Diffusion Policy Visuomotor Policy Learning via Action Diffusion RSS 2023

---

### Official Review · Reviewer_8bw9 · 2025-03-14

**Overall Recommendation:** 2

**Summary:**

The authors propose combining the reasoning capabilities of LLMs with the robot action generalization capabilities of diffusion models, creating DiVLA. DiVLA extracts and interleaves tokens from visual input and text using SigLIP, concatenates them, and processes them in a VLM. The VLM generates action tokens that are projected and processed by the diffusion model. FiLM layers are used to incorporate reasoning tokens output by the VLM into the diffusion model. DiVLA is compared to previous VLAs on two real-world robot settings for various pick and place tasks, including seen and unseen objects and including a bimanual robot.

**Claims And Evidence:**

Since the paper makes the claim that reasoning contributes to the model’s robustness in l. 365-368 col. 2 then this should be validated.

**Essential References Not Discussed:**

N/A

**Experimental Designs Or Analyses:**

Yes.

**Methods And Evaluation Criteria:**

Yes but there are no synthetic benchmarks, implying that the model may not perform well on an environment with a vastly different set-up than the real-world experiments currently used.

**Other Comments Or Suggestions:**

- l. 306 col. 1 “pretraiend”→”pretrained”
- l. 309 col. 1 “the setting as pi_0”→”pi_0’s setting”

**Other Strengths And Weaknesses:**

Strengths:

S1: Strong performance on two real world settings, especially on generalization to unseen objects.

Weaknesses:

W1: The components of DiVLA are not ablated. Since the paper makes the claim that reasoning contributes to the model’s robustness in l. 365-368 col. 2 then this should be validated. Additionally it is important to validate the choice of model architectures, since the main contribution of DiVLA is in the combination of existing components.

W2: No synthetic benchmarks, such as Libero [1], that are standard for pick and place tasks in robotics. This implies that the model may not perform well on an environment with a vastly different set-up than the real-world experiments currently used.

[1] LIBERO: Benchmarking Knowledge Transfer for Lifelong Robot Learning. Liu et al. arxiv preprint arxiv:2306.03310, 2023.

**Questions For Authors:**

1. What is the difference between points 1 and 6 in the introduction section?
2. l. 140-144 “Research has shown … various embodiments.” Can you give some citations with evidence for this?
3. What are problems with current diffusion-based VLAs?
4. l. 248-252 col. 2 “To … reasoning.” What prompts do you send to GPT-4o?
5. l. 364 - 376 Does this pose a problem when there are two or more similar objects in the scene, i.e. does the model confuse similar objects when both exist in the scene?
6. How does the DiVLA’s size compare to other VLAs such as Diffusion Policy, Octo, OpenVLA etc? How long does it take to infer a single image and to train compared to the other models?
7. What is the difference between reasoning and action tokens? How is the VLM trained to output them separately? Is there a different number of each type of token generated?

**Relation To Broader Scientific Literature:**

DiVLA builds directly off of Diffusion Policy and previous work in VLMs. It tackles a worthwhile problem of combining the best of VLMs and diffusion models for VLAs.

**Theoretical Claims:**

N/A

---

> ### Author Rebuttal · Authors · 2025-04-01
>
> Thank you for your thorough review and valuable feedback.
> ## 1. Simulated evaluation
> Real-world evaluation is more challenging than simulation. While our work emphasizes complex real-world tasks like long-horizon bin-picking and bimanual table bussing, we also evaluate DiVLA on two standard simulation benchmarks, Calvin and Libero. Compared with baselines including Diffusion Policy, Octo, OpenVLA, RT-1, Robo-Flamingo, and GR-1, DiVLA achieves the best performance under a unified training setup (results: https://i.postimg.cc/xTybS5p3/simulation.png), demonstrating its robustness across both real and simulated environments.
>
> ## 2. Ablation on key modules
>
> We do agree that ablation study is necessary for better support our claim. We have addressed this problem
> in response to **Reviewer fKzV#3**. Please refer to that section for details.
>
> ## 3. Clarify difference between points 1 and 6 in the introduction section
>
> We are sorry for the confusion between point 1 and point 6. Generalizing manipulation skills on novel objects and robust to dynamic environment are two main challenges in robot learning. Point 1 focuses on DiVLA’s ability to recognize novel objects through self-generated reasoning, while Point 6 highlights its robustness to distractors and dynamic environments.
>
> ## 4. Citation for “Research has shown … various embodiments.”
>
> Thanks for pointing this out. The next-token-prediction method used in OpenVLA struggles to complete tasks when adapting to new embodiments [1] and performs poorly when learning dexterous skills with high-frequency control [2].
>
> ## 5. Limitation of current diffusion-based VLAs and Superiority of DiVLA
>
> Thanks for your insightful question. We have addressed this problem
> in response to **Reviewer fKzV#5**. Please refer to that section for details.
>
> ## 6. Prompt for gpt-4o
>
> Thanks for pointing this out. The prompt is at here: https://i.postimg.cc/kXcDH4Dj/prompt.jpg.
>
> ## 7. Clarify whether the model will confuse similar objects or not
>
> Thanks for your reminder. While DiVLA shows strong generalization, it occasionally misclassifies unseen objects, such as confusing small stuffed toys with toy cars. However, DiVLA handles shape-similar, color-distinct objects well, thanks to its robust color perception. Real-time reasoning traces help diagnose misclassifications and offer insights for further improvement.
>
> ## 8. Comparison on model size, inference speed, training speed with baselines
>
> Thanks for pointing this out. We will discuss the model size and inference speed separately.
>
> 1). **Model size:** Our DiVLA, which uses only 2B parameters—1/3 of OpenVLA’s parameters—achieves the best performance across all tasks, demonstrating our method's model efficiency.
>
> | model | DP | Octo | OpenVLA | DiVLA-2B |
> | --- | --- | --- | --- | --- |
> | size | 153M | 93M | 7B | 2B |
>
> 2). **Inference speed:** As shown in table, DiVLA-2B can achieve 82hz control frequency at test time with vLLM and action chunk. This is 20x faster than OpenVLA. The results highlight the inference efficiency of diffusion-based VLAs over autoregressive VLAs.
>
> | model | DP | Octo | OpenVLA | DiVLA-2B |
> | --- | --- | --- | --- | --- |
> | inference speed(Hz) | 122 | 105 | 4 | 82 |
>
> 3). **Training speed:** We have addressed this problem in response to **Reviewer fKzV#1**. Please refer to that section for details.
>
> ## 9. Clarify details on reasoning and action tokens
>
> Thanks for your feedback.
>
> 1). **Difference between reasoning and action tokens:** Action and reasoning tokens serve as different conditions on action denoising. Specifically, action tokens mainly include visual embeddings providing global observations and raw instruction tokens. On the other hand, reasoning tokens encapsulate hierarchical substep-level features that decompose complex tasks into temporally executable steps.
>
> 2). **How is VLM trained to output reasoning and action tokens:** Action tokens refer to vanilla vision tokens and tokens from raw instruction processed by VLM backbone while reasoning tokens are task-specific language generated by VLM. Thus, we only use standard next-token prediction loss for supervising reasoning generation.
>
> 3). **Number of both tokens:** As previously noted, action tokens comprise tokens from raw instruction and visual tokens, whereas reasoning tokens are autonomously generated by the VLM. Action tokens are much longer than reasoning tokens as a large number of vision tokens.
>
> ## 10. Typo problems
> Thanks for your feedback. We will improve them in updated version.
>
> [1]. TinyVLA: Towards Fast, Data-Efficient Vision-Language-Action Models for Robotic Manipulation
>
> [2]. FAST: Efficient Action Tokenization for Vision-Language-Action Models
>
> [3]. CogACT: A Foundational Vision-Language-Action Model for Synergizing Cognition and Action in Robotic Manipulation
>
> [4]. RDT-1B: a Diffusion Foundation Model for Bimanual Manipulation
>
> [5]. π0: A Vision-Language-Action Flow Model for General Robot Control

---

### Official Review · Reviewer_fKzV · 2025-03-17

**Overall Recommendation:** 3

**Summary:**

DiffusionVLA unifies autoregressive reasoning with diffusion-based action policies to build robust vision–language–action models for robotic control. By injecting self-generated reasoning directly into the policy head, the framework improves interpretability and decision-making. Extensive experiments on tasks like factory sorting and zero-shot bin picking demonstrate strong generalization and fast inference, outperforming several state-of-the-art baselines.

**Claims And Evidence:**

-  The paper claims that combining autoregressive reasoning with diffusion policies and a dedicated reasoning injection module yields superior and interpretable robotic control compared with prior VLA models.

While the empirical results show improved task performance and interpretability, similar capabilities (e.g., natural language rationale generation) have been explored in comparable works. A more detailed ablation or comparison that isolates the effect of the reasoning module would strengthen the claim (e.g., the choice of projector, the effect of data source).

**Essential References Not Discussed:**

The reference list is comprehensive and appropriately covers related work.

**Experimental Designs Or Analyses:**

- Projection Layer for Action Tokens:

Have the authors experimented with different projection layers, such as FiLM, instead of the two-MLP+LayerNorm design? A comparison would help justify the module selection.

- Reasoning Injection Module:

Similarly, for the reasoning injection module, did the authors try simpler alternatives (e.g., a plain MLP or an attention-based mechanism) and compare their effectiveness? How do the authors guarantee that the chosen design is optimal?

- Learning Rate Details:

Is the fixed learning rate (2e-5) uniformly applied to all layers, including the newly initialized projection modules?

**Methods And Evaluation Criteria:**

- Training Efficiency and Model Size:

How does DiffusionVLA’s training efficiency and model size compare against baseline methods? Quantitative comparisons (e.g., training time, parameter counts, memory usage) would provide valuable context.

**Other Comments Or Suggestions:**

- Add a period after “randomly initialized weights” for clarity.
- In Figure 1, annotate the x-axis to indicate model size.
- Consider renaming Section 3.2 from “Model Design Choices” to a title that more accurately reflects its focus on architecture and training objectives.
- Clarify whether the fixed learning rate (2e-5) is applied uniformly across all layers, including newly initialized projection modules.
- The claim that the model “generates natural language rationales alongside its output actions” is interesting; please discuss if any baseline models offer similar interpretability features and how DiffusionVLA’s approach compares.

**Other Strengths And Weaknesses:**

### Strengths:

- The unified framework effectively combines reasoning and diffusion-based action generation.
- The reasoning injection module adds interpretability by generating natural language rationales alongside actions.
- Empirical results show robust performance across varied real-world tasks with fast inference speeds.

### Weaknesses:

- Limited quantitative discussion on training efficiency and model size compared with baseline methods.
- The paper does not provide comparative experiments on alternative projection or reasoning modules (e.g., using FiLM or plain MLPs).

**Questions For Authors:**

See the comments and questions above.

**Relation To Broader Scientific Literature:**

A deeper discussion is needed on how this work situates itself within the broader literature on diffusion policies and autoregressive reasoning in robotics. How do its capabilities (especially regarding interpretability and efficiency) compare with models like OpenVLA, or π0?

**Theoretical Claims:**

There are no novel theoretical contributions.

---

> ### Author Rebuttal · Authors · 2025-04-01
>
> Thanks for your careful review and valuable comments. We address each question below.
>
> ## 1. Training efficiency for DiVLA compared to baselines
>
> Thank you for your valuable feedback and insightful advice. Since DP and Octo perform significantly worse across most tasks, our comparison focuses on competitive VLAs.
>
> 1). **Pretraining Computations:** OpenVLA requires 27 epochs (21,500 A100 GPU hours) for action token prediction on the OXE dataset, while DiVLA achieves language rationale generation in just 2.5 epochs (155 H800 GPU hours) on part of Droid dataset. DiVLA uses only 39K pretraining samples, 25 times fewer than OpenVLA's 970K, yet still delivers interpretable reasoning, demonstrating superior efficiency.
>
> | model | Pre-trained Data | GPU hours |
> | --- | --- | --- |
> | TinyVLA | / | / |
> | OpenVLA | 970K | 21500 |
> | DiVLA-2B | 39K | 155 |
>
> 2). **Finetuning Computations:** All models are trained with the same batch size on 8×H800 GPUs. TinyVLA takes 12 hours to adapt, while DiVLA outperforms it by 29.8% in factory sorting and reduces finetuning time compared to OpenVLA, achieving a 20.9% improvement. DiVLA shows superior efficiency and adaptability in finetuning.
>
> | model | OpenVLA | TinyVLA | DiVLA-2B |
> | --- | --- | --- | --- |
> | Finetune Time (hours) | 25 | 12 | 15 |
> | size | 7B | 1.3B | 2B |
> | GPU memory | 66 | 29 | 38 |
>
> 3). **Model Size and Memory Usage:** As shown in the table, DiVLA is smaller than OpenVLA (2B vs. 7B parameters) and requires less GPU memory, making it more reproducible and accessible.
>
> Overall, DiVLA demonstrates superior computational efficiency in both pretraining and finetuning, achieving significant performance gains and highlighting the advantage of our approach.
>
> ## 2. Ablation on projection layer of action tokens
>
> Thanks for pointing it out. We ablate two projection layer, as requested for action tokens on 3 tasks (described in page 4 line 189-199). For each variants we evaluated them for 11 trails for each task.
>
> | task/module | FiLM | two-MLP+LayerNorm(ours) |
> | --- | --- | --- |
> | task 1 | 63.7 | 100 |
> | task 2 | 45.4 | 100 |
> | task 3 | 36.4 | 63.6 |
> | Average | 48.5 | 87.9 |
>
> As shown in table, using two-MLP+LayerNorm as projection layer performs relatively better than FiLM which highlights our choice is effective.
>
> ## 3. Ablation on three reasoning injection modules (MLP, Q-Former, and FiLM)
>
> We thank the reviewer so much for pointing this out. We ablate three reasoning injection modules, 1) Plain MLP like LLaVA [1], 2) Q-former as in BLIP [2], and 3) FiLM. For each variants, we evaluated on three tasks, each with 11 trials.
>
> | task/module | Plain MLP | Q-Former | FiLM(ours) |
> | --- | --- | --- | --- |
> | Task 1 | 36.4 | 18.2 | 100 |
> | Task 2 | 45.4 | 45.4 | 100 |
> | Task 3 | 18.2 | 27.3 | 63.6 |
> | Average | 33.3 | 30.1 | 87.9 |
>
> As demonstrated in the table, DiVLA with the FiLM module significantly outperforms plain MLP and Q-former, confirming the efficacy of our design. Intuitively, reasoning tokens act as conditioning factors, enhancing robot action generalization without dominating predictions. The FiLM architecture effectively fulfills this conditioning role.
>
> ## 4. Do we use 2e-5 for all module?
>
> Yes, we use initial learning rate of 2e-5 to all layers, including newly initialized projection modules. We didn’t do hyper-parameter search in our experiments.
>
> ## 5. It is interest that Diffusion-VLA can generate language rationales alongside its robot action, how does it compare to OpenVLA and pi0 in terms of  interpretability
>
> Thank you for raising this point. DiVLA is the only model that generates language rationales alongside robot actions, making it interpretable. In contrast, OpenVLA and Pi0 cannot generate language. DiVLA uses reasoning for dynamic scene understanding and task planning, revealing the model's decision-making process and improving long-horizon task completion by making failure reasons visible to users.
>
> Our experiments shows the importance of making VLA interpretable. Specifically, as illustrated in Figure 3, on the sorting task involving 4 progressively challenging settings, DiVLA can generate real-time reasoning traces (e.g., identifying objects and categorizing them). It achieves 20.9% improvement over OpenVLA across all settings. Furthermore, as shown in Figure 4, DiVLA exhibits remarkable zero-shot generalization, outperforming OpenVLA by 35.3% on 102 unseen objects. These experimental results shows that introducing interpretability to VLAs significantly enhances their ability to execute long-horizon tasks and generalize to novel objects.
>
> ## 6. For format, section renaming and figure annotation
>
> Thanks for pointing this out. We will improve these formating problems, section renaming and figure annotation in updated version.
>
> [1]. Improved Baselines with Visual Instruction Tuning
>
> [2]. BLIP: Bootstrapping Language-Image Pre-training for Unified Vision-Language Understanding and Generation

---

### Decision · Program_Chairs · 2025-05-01

**Decision:**

Accept (poster)

**Comment:**

In this paper, the authors propose DiffusionVLA (DiVLA), a hybrid model integrating VLM-based reasoning with diffusion-based action generation for robotics via a reasoning injection module (FiLM layers). The core idea involves using a VLM to generate explicit reasoning tokens (natural language rationales) based on visual input and instructions, which are then injected into a diffusion-based policy head via a FiLM module to condition the action generation process. The claim is that autoregressive action generators often lack the level of precision required for robotics, while diffusion models do better; but on the flip side, diffusion models are not good at reasoning. This approach is then tested on several real-world tasks such as pick and place, sorting, bin picking etc., showing significant improvements over baselines such as Diffusion Policy, OpenVLA among others.

Initial reviews were mixed, and towards the end still leaned negative (one weak accept, three weak rejects). Strengths identified by the reviewers included the core idea, strong real-world/zero-shot results, and interpretability via rationales. There were also some questions regarding the novelty of the interpretability feature, and concerns about lack of ablations/comparisons within this context. But the major weaknesses that were pointed out were the lack of crucial ablations and standard synthetic benchmarks.

However, the authors provided a detailed and effective rebuttal that substantially addressed these key concerns. To summarize, the rebuttal addressed the following:

1. Missing Ablation Studies: Authors provided quantitative ablation results comparing different choices for the action token projection layer (showing their MLP+LayerNorm outperformed FiLM) and the reasoning injection module (showing their FiLM choice significantly outperformed Plain MLP and Q-Former), justifying their design choices (addressing R-fKzV#2, R-fKzV#3, R-8bw9#2).
2. Lack of Simulation Benchmarks: Included new results demonstrating state-of-the-art performance on standard simulation benchmarks (CALVIN/LIBERO), broadening the evaluation beyond the initial real-world tasks (addressing R-8bw9#1).
3. Training Efficiency & Model Size: Provided tables comparing DiVLA's pre-training requirements (significantly fewer GPU hours and data than OpenVLA), fine-tuning time (competitive), model size (smaller than OpenVLA), and GPU memory usage against key baselines like OpenVLA and TinyVLA (addressing R-fKzV#1, R-8bw9#8).
4. Inference Speed Justification: Offered detailed explanations differentiating diffusion-based action generation (fewer steps for sequences) from autoregressive methods (token-per-DoF). Provided quantitative breakdowns of inference time with and without vLLM acceleration, clarifying the sources of speedup and quantifying vLLM's impact. Explained the action chunking strategy used (addressing R-13rJ#1, R-bZhD#1).
5. Clarification on Reasoning/Action Tokens: Detailed the distinction between action tokens (visual/instruction embeddings) and reasoning tokens (generated language steps), how each is generated (backbone processing vs. next-token prediction), their relative counts, and confirmed both serve as conditioning inputs for the diffusion model (addressing R-8bw9#7, R-13rJ#2). Provided a code link for reasoning generation.

Authors also addressed generalization vs. memory concern and other methodological details. While the initial submission had gaps, I believe that the rebuttal successfully addressed several of them, strengthening the paper's claims regarding performance, implementation details, particularly zero-shot generalization, and efficiency. Some of the subsequent concerns raised by reviewer 13rJ were also addressed by the authors in a followup comment, but the discussion was not entirely finalized, and I do believe the authors addressed the reviewer's concerns in that thread satisfactorily.

The core contribution of integrating explicit reasoning into diffusion policies is sound and now better supported empirically. Given the combination of the novel approach, strong demonstrated performance across varied tasks (real-world and simulation), and the valuable interpretability feature - all bolstered by a fairly strong rebuttal and the willingness of the authors to elaborate on details of their technique; pushes the paper towards acceptance. Therefore, despite initial reservations, I recommend this paper to be accepted, contingent on careful revision to incorporate the extensive new results and clarifications in the final version of the paper.